# Ensemble sampling for linear bandits:
# small ensembles suffice

**David Janz**
University of Oxford[*]
david.janz@stats.ox.ac.uk

**Alexander E. Litvak**
University of Alberta
alitvak@ualberta.ca

**Csaba Szepesvári**
University of Alberta
szepesva@ualberta.ca

## Abstract

We provide the first useful and rigorous analysis of ensemble sampling for the stochastic linear bandit setting. In particular, we show that, under standard assumptions, for a $d$-dimensional stochastic linear bandit with an interaction horizon $T$, ensemble sampling with an ensemble of size of order $d \log T$ incurs regret at most of the order $(d \log T)^{5/2}\sqrt{T}$. Ours is the first result in any structured setting not to require the size of the ensemble to scale linearly with $T$—which defeats the purpose of ensemble sampling—while obtaining near $\sqrt{T}$ order regret. Our result is also the first to allow for infinite action sets.

## 1 Introduction

Ensemble sampling (Lu and Van Roy, 2017) is a family of randomised algorithms for balancing exploration and exploitation within sequential decision-making tasks. These algorithms maintain an ensemble of perturbed models of the value of the available actions and, in each step, select an action that has the highest value according to a model chosen uniformly at random from the ensemble. The ensemble is then incrementally updated the new observations.

Ensemble sampling was introduced as an alternative to Thompson sampling (Thompson, 1933) that is tractable whenever incremental model updates are cheap (Osband et al., 2016; Lu and Van Roy, 2017). It is thus particularly popular in deep reinforcement learning, where the models (neural networks) are trained via gradient descent in an incremental fashion. Therein, ensemble sampling features directly in algorithms such as *bootstrapped DQN* and *ensemble+* (Osband et al., 2016, 2018), as part of other methods (Dimakopoulou and Van Roy, 2018; Curi et al., 2020), and as the motivation for methods, including *random network distillation* (Burda et al., 2018) and *hypermodels for exploration* (Dwaracherla et al., 2020). Ensemble sampling has also been applied to online recommendation systems (Lu et al., 2018; Hao et al., 2020; Zhu and Van Roy, 2021), the behavioural sciences (Eckles and Kaptein, 2019) and to marketing (Yang et al., 2020).

Despite its practicality and simple nature, ensemble sampling has thus far resisted analysis. Indeed, Qin et al. (2022), who showed a bound of order $\sqrt{T}$ on its *Bayesian regret* under the impractical condition that the size of the ensemble scales at least linearly with the horizon $T$, summarise the state-of-the-art in this regard as follows:

> 'A lot of work has attempted to analyze ensemble sampling, but none of them has been successful.'

Readers familiar with the literature know that the analysis of randomised exploration methods, such as Thompson sampling, or perturbed history exploration (Kveton et al., 2020; Janz et al., 2024), relies on showing that at each step there is a constant probability of choosing a model that overestimates the true value (Agrawal and Goyal, 2013; Abeille and Lazaric, 2017). For these methods, the analysis

---

[*]Work carried out while David Janz was a postdoc with Csaba Szepesvári at the University of Alberta.

38th Conference on Neural Information Processing Systems (NeurIPS 2024).

is relatively simple, because the algorithms can be described as first fitting a model to the past data, and then using a fresh source of randomness to perturb the model, which is then used to derive the action to be used. Ensemble sampling does not fit this pattern: here, the distribution of models given the past is controlled only implicitly, making the analysis challenging.

Our contribution is a guarantee that (a symmetrised version of) ensemble sampling, given an ensemble size logarithmic in $T$ and linear in the number of features $d$, incurs regret no worse than order $(d \log T)^{5/2} \sqrt{T}$. This is the first successful analysis of ensemble sampling in the stochastic linear bandit setting, or indeed, any structured setting (see Remarks 6 and 7 for a discussion of this claim). Our result is based on (a slight extension of) the now-standard framework of Abeille and Lazaric (2017), and as such it ought to be possible to extend it to the usual settings: generalised linear bandits (Filippi et al., 2010), kernelised bandits (Srinivas et al., 2010), deep learning (via the neural tangent kernel, per Jacot et al., 2018) and reinforcement learning (following the work of Zanette et al., 2020).

## 2 Linear ensemble sampling

We now outline our problem setting, a version of the linear ensemble sampling algorithm and our main result, and discuss these in the context of prior literature. We encourage readers less familiar with the motivation around ensemble sampling to consult Lu and Van Roy (2017) or Osband et al. (2019); our focus will be on analysis. We will use the following notation:

**Sets and real numbers** We write $\mathbb{N}^+ = \{1, 2, \dots\}$, $\mathbb{N} = \mathbb{N}^+ \cup \{0\}$ and $[n] = \{1, 2, \dots, n\}$. We use the shorthand $(a_t)_t$ for a sequence indexed by $t$ in $\mathbb{N}$ or $\mathbb{N}^+$ when this index set can be deduced from the context. For $a, b \in \mathbb{R}$, that is, for real numbers $a$ and $b$, $a \wedge b$ denotes their minimum and $a \vee b$ their maximum.

**Vectors and matrices** We identify vectors and linear operators with matrices. We write $\| \cdot \|_2$ for the 2-norm of a vector, and $\| \cdot \|$ for the operator norm of a matrix corresponding to the largest singular value, which we denote by $s_1(\cdot)$; we write $s_2(\cdot), s_3(\cdot), \dots$ for the remaining singular values, ordered descendingly. For vectors $u, v$ of matching dimension, $\langle u, v \rangle = u^\mathsf{T} v$ denotes their usual inner product. We write $\mathbb{B}_2^d$ for the 2-ball in $\mathbb{R}^d$, $\mathbb{S}_2^{d-1} = \partial \mathbb{B}_2^d$ for its surface, the $(d-1)$-sphere, and $H_u$ for the closed half-space $\{v \in \mathbb{R}^d : \langle u, v \rangle \geq 1\}$.

**Probability** We work on a probability space with probability measure $\mathbb{P}$, and denote the corresponding expectation operator by $\mathbb{E}$. We write $\sigma(\cdot)$ for the $\sigma$-algebra generated by the argument. For an event $A$, we write $\mathbf{1}[A]$ for the indicator function of $A$.

**Uniform distribution** We write $\mathcal{U}(B)$ for the uniform distribution over a set $B$ (when well-defined), and $\mathcal{U}(B)^{\otimes m}$ for its $m$-fold outer product; that is, $(U_1, \dots, U_m) \sim \mathcal{U}(B)^{\otimes m}$ are $m$ i.i.d. random variables with common law $\mathcal{U}(B)$.

### 2.1 Problem setting: stochastic linear bandits

We consider the standard stochastic linear bandit setting. At each step $t \in \mathbb{N}^+$, a learner selects an action $X_t$ from an action set $\mathcal{X}$, a compact subset of $\mathbb{B}_2^d$, and receives a random reward $Y_t \in \mathbb{R}$ that obeys the following assumption:

**Assumption 1.** *At each time step $t \in \mathbb{N}^+$, the agent receives a reward of the form $Y_t = \langle X_t, \theta^\star \rangle + Z_t$ with $\theta^\star \in \mathbb{B}_2^d$ a fixed instance parameter and $Z_t$ a random variable satisfying*

$$\mathbb{E}[\exp\{s Z_t\} \mid \sigma(\mathcal{F}_{t-1} \cup \mathcal{A}_t \cup \sigma(X_t))] \leq \exp\{s^2/2\}, \quad \forall s \in \mathbb{R},$$

*almost surely, where $\mathcal{F}_{t-1} = \sigma(X_1, Y_1, \dots, X_{t-1}, Y_{t-1})$ and $\mathcal{A}_t$ is the $\sigma$-algebra generated by the random variables used by the learner to select its actions up to and including time $t$.*

The learner is given a horizon $T \in \mathbb{N}^+$ and its performance is evaluated by the (pseudo-)regret, $R(T)$, incurred for the first $T$ steps, defined as

$$R(T) = \max_{x \in \mathcal{X}} \sum_{t=1}^{T} \langle x - X_t, \theta^\star \rangle,$$

The smaller the regret, the better the learner. Under our assumptions, up to logarithmic factors, the regret of the best learners is of order $d\sqrt{T}$ (Chapters 19–24, Lattimore and Szepesvári, 2020). Our

main result will show that if a learner follows the upcoming ensemble sampling algorithm to select the actions $X_1, \ldots, X_T$, the regret it incurs will satisfy a similar high probability bound, with a slightly worse dependence on the dimension $d$.

## 2.2 Algorithm: linear ensemble sampling

Linear ensemble sampling, listed as Algorithm 1, proceeds as follows. At the outset, we fix an ensemble size $m \in \mathbb{N}^+$, a regularisation parameter $\lambda > 0$ and a sequence of perturbation scale parameters $r_0, r_1, r_2, \ldots > 0$. Then, before the start of each round $t \in \mathbb{N}^+$, linear ensemble sampling computes $m + 1$ $d$-dimensional vectors. The first of these is the usual ridge regression estimate of $\theta^\star$,

$$\hat{\theta}_{t-1} = V_{t-1}^{-1} \sum_{i=1}^{t-1} X_i Y_i \quad \text{where} \quad V_{t-1} = V_0 + \sum_{i=1}^{t-1} X_i X_i^\mathsf{T} \quad \text{and} \quad V_0 = \lambda I. \tag{1}$$

The remaining $m$ parameter vectors, which shall serve as perturbations, are of the form

$$\tilde{\theta}_{t-1}^j = V_{t-1}^{-1} \left( S_0^j + \sum_{i=1}^{t-1} X_i U_i^j \right) \quad \text{for each} \quad j \in [m],$$

where $S_0^j \in \mathbb{R}^d$ is a *random initialisation*, and is taken to be uniform on $\lambda \sqrt{d} \mathbb{S}_2^{d-1}$, and $U_1^j, U_2^j, \ldots$ are *random targets*, uniform on the interval $[-1, 1]$. All of these random variables are independent of the past at the time they are sampled and across the $m$-many replications. The algorithm then selects a random index $J_t \in [m]$ and sign $\xi_t \in \{\pm 1\}$, both uniformly distributed over their respective ranges, computes the perturbed parameter

$$\theta_t = \hat{\theta}_{t-1} + r_{t-1} \xi_t \tilde{\theta}_{t-1}^{J_t}$$

and selects an action

$$X_t \in \arg\max_{x \in \mathcal{X}} \langle x, \theta_t \rangle,$$

with ties dealt with in a measurable way. All in all, the vectors

$$\hat{\theta}_{t-1} \pm r_{t-1} \tilde{\theta}_{t-1}^1, \ldots, \hat{\theta}_{t-1} \pm r_{t-1} \tilde{\theta}_{t-1}^m$$

serve as an ensemble of $2m$-many perturbed estimates of $\theta^\star$, and in each round the algorithm acts optimally with respect to one of these, selected at uniformly at random.

Our linear ensemble sampling algorithm deviates in two ways from that of Lu and Van Roy (2017):

1. The random sequence of targets used to fit the ensembles in our algorithm consists of uniform random variables rather than the Gaussians used in the previous literature. This choice simplifies the proofs, but our results hold (up to constant factors) for any suitable subgaussian targets, including Gaussian—see Remark 14 for a sketch of the argument.

2. We symmetrise our perturbations using Rademacher random variables, which does not feature in previous formulations of the ensemble sampling algorithm. This helps to create a more uniform distribution of the perturbations $(\tilde{\theta}_t^j)_t$ around zero with minimal computational overhead, and is important for our proof technique.

In Appendix A, we provide reformulation of our linear ensemble sampling algorithm that is more in line with the style of presentation of the algorithm given by Lu and Van Roy (2017).

## 2.3 Regret bound for linear ensemble sampling

Our advertised result is captured in the following theorem. To state the theorem we need the sequence

$$\beta_t^\delta = \sqrt{\lambda} + \sqrt{2 \log(1/\delta) + \log(\det(V_t)/\lambda^d)}, \quad t \in \mathbb{N}.$$

Here and in related quantities, the superscripted $\delta$ expresses the dependence on a $\delta \in (0, 1]$.

---
**Algorithm 1** Linear ensemble sampling

---
**Input** regularisation parameter $\lambda > 0$, ensemble size $m \in \mathbb{N}^+$, perturbation scales $r_0, r_1, \ldots > 0$
Sample $(S_0^j)_{j \in [m]} \sim \mathcal{U}(\lambda \sqrt{d} \mathbb{S}_2^{d-1})^{\otimes m}$
Let $V_0 = \lambda I$, $\hat{\theta}_0 = 0$, and let $\tilde{\theta}_0^j = V_0^{-1} S_0^j$ for each $j \in [m]$
**for** $t \in \mathbb{N}^+$ **do**
    Sample $(\xi_t, J_t) \sim \mathcal{U}(\{\pm 1\} \times [m])$ and let $\theta_t = \hat{\theta}_{t-1} + r_{t-1} \xi_t \tilde{\theta}_{t-1}^{J_t}$
    Compute an $X_t \in \arg\max_{x \in \mathcal{X}} \langle x, \theta_t \rangle$, play action $X_t$ and receive reward $Y_t$
    Sample $(U_t^j)_{j \in [m]} \sim \mathcal{U}([-1, 1])^{\otimes m}$ and let $S_t^j = S_{t-1}^j + U_t^j X_t$ for each $j \in [m]$
    Let $V_t = V_{t-1} + X_t X_t^\mathsf{T}$, $\hat{\theta}_t = V_t^{-1} \sum_{i=1}^t Y_i X_i$, and let $\tilde{\theta}_t^j = V_t^{-1} S_t^j$ for each $j \in [m]$

---

**Theorem 1.** *Fix $\delta \in (0, 1]$ and take $r_t = 7\beta_t^\delta$ for all $t \in \mathbb{N}$, $\lambda \geq 5$ and $m \geq 400 \log(NT/\delta)$ for $N = (134\sqrt{1 + T/\lambda})^d$. Then there exists a universal constant $C > 0$ such that, with probability at least $1 - \delta$, the regret incurred by a learner following linear ensemble sampling with these parameters in our stochastic linear bandit setting (formalised in Assumption 1) is bounded as*

$$R(\tau) \leq Cm^{3/2}\beta_{\tau-1}^\delta \left( \sqrt{d\tau \log(1 + \tau/(\lambda d))} + \sqrt{(\tau/\lambda)\log(\tau/(\lambda\delta))} \right) \quad \text{for all} \quad \tau \in [T].$$

**Remark 1.** *If $\delta \geq 1/T^\alpha$ for some $\alpha > 0$ and we take $\lambda = 5$ and $m$ to be as small as possible given the constraint of Theorem 1, then*

$$m \leq C_\alpha d \log T \quad \text{and} \quad R(T) \leq C_\alpha' (d \log T)^{5/2} \sqrt{T}$$

*for some constants $C_\alpha, C_\alpha' > 0$ that depend on $\alpha$ only and where the bound on the regret holds with probability $1 - \delta$. That is, for polynomially sized confidence parameters, the ensemble size scales linearly with $d \log(T)$, while the regret scales with $d^{5/2}\sqrt{T}$ up to logarithmic-in-T factors. The latter scaling is slightly worse than that obtained for Thompson sampling (cf. Theorem 17), where the regret scales with $d^{3/2}\sqrt{T}$, again, up to logarithmic-in-T factors.*

**Remark 2.** *Theorem 1 does not recover the expected behaviour for large ensemble sizes; that is, as $m \to \infty$. On one hand, this is not an issue: we are interested in the practical regime where the ensemble size is small. On the other, it suggests that better analysis might be possible. In Remark 12, we discuss a potential looseness in our analysis, which, if addressed, would result in the removal of a factor of $m$ from the bound, thus bringing the regret in line with that of Thompson sampling. The technique that would be required to do so could then also be used to change the remaining $\sqrt{m}$ dependence to an order $\sqrt{d \log T/\delta}$ dependence, thus recovering a bound with the right dependence on $m$. However, as discussed in Remark 12, such improvements, if possible, may be hard to attain.*

**Remark 3.** *Our ensemble sampling algorithm requires the ensemble size $m$ to be fixed in advance. As $m$ depends on the horizon $T$, the method only provides guarantees for a fixed, finite horizon $T$, and our regret bound has a direct dependence on $T$. The doubling trick would be a simple, but somewhat unsatisfactory way of removing this dependence. An alternative is to grow the ensemble online. However, a naive implementation of growing the ensemble size would require storing all past observations and computation per time step also growing with time, which is counter to the idea of a fast incremental method.*

**Remark 4.** *In light of Theorem 1, linear ensemble sampling might be seen as a less effective and less computationally efficient version of linear Thompson sampling. This is correct: in the linear setting, where Thompson sampling may be implemented in closed form, ensemble sampling has no clear advantages. With that said, variants of ensemble sampling and related bootstrapping methods are popular in more complex settings where closed forms are not available (say, deep reinforcement learning, per Osband et al., 2016, 2018), and the linear setting provides an important testing ground for the soundness of these algorithms.*

**Remark 5.** *Instead of using a randomly sampled sign and index $(\xi_t, J_t)$, we could use the upper-confidence-bound strategy of selecting the actions using a maximum over the ensemble, that is*

$$X_t \in \arg\max_{x \in \mathcal{X}} Q(x) \quad \text{for} \quad Q(x) = \max\{\langle x, \hat{\theta}_{t-1} + r_{t-1}\xi^\star \tilde{\theta}_{t-1}^{J^\star}\rangle : (\xi^\star, J^\star) \in \{\pm 1\} \times [m]\}.$$

*It is immediate from the proof of Theorem 1 that this would yield a regret bound scaling with $d^{3/2}$. This raises the question of whether this max-over-ensemble approach is actually superior, or whether it's just that its much simpler analysis more readily yields a good bound.*

## 2.4 Comparison to related results

We now discuss the results of Lu and Van Roy (2017), Qin et al. (2022), and Lee and Oh (2024), showing that our result is the first to begin justifying the practical effectiveness of ensemble sampling.

**Remark 6.** *The work of Lu and Van Roy (2017) makes strong claims on the frequentist regret of linear ensemble sampling. However, as pointed out by Qin et al. (2022), and confirmed by Lu and Van Roy (2017) in their updated arxiv manuscript, the analysis of Lu and Van Roy (2017) is flawed.*

**Remark 7.** *The only correct result on the regret of linear ensemble sampling is by Qin et al. (2022), which gives that for a $d$-dimensional linear bandit with an action set $\mathcal{X}$ of cardinality $K$, the Bayesian regret incurred—that is, average regret for $\theta^\star \sim \mathcal{N}(0, I_d)$—is bounded as*

$$BR(T) \leq C\sqrt{dT \log K} + CT\sqrt{\frac{K \log(mT)}{m}}(d \wedge \log K)$$

*for some universal constant $C > 0$. Observe that this bound needs an ensemble size linear in $T$ for Bayesian regret that scales as $\sqrt{T}$ (up to constant and polylogarithmic factors), which largely defeats the purpose of ensemble sampling. Furthermore, the ensemble size $m$ needs to scale linearly with $K$ to get a $\log K$ overall dependence on $K$. Therefore, to tackle a bandit with $\mathcal{X} = \mathbb{B}_2^d$ using the bound of Qin et al. (2022) and discretisation, since order $2^{d-1}$-many actions would be needed to guarantee a small discretisation error, the ensemble size $m$ would need to scale exponentially in $d$.*

**Remark 8.** *Building on our result, Lee and Oh (2024) have shown that for a $K$-armed linear bandit, an ensemble of size on the order of $K \log T$ suffices to ensure a bound on $R(T)$ on the order of $d^{3/2}\sqrt{T}$, ignoring logarithmic-in-$T$ factors. This gives a regret bound tighter by a factor of $d$ relative to ours, but at the cost of replacing the dimension $d$ by the number of arms $K$ in the ensemble size. Recalling that, $K$ can be exponential in $d$ (per Remark 12), their result may require rather large ensemble sizes. Lee and Oh (2024) also contribute some rather curious remarks about our proofs.*

## 3 Analysis of linear ensemble sampling

Our analysis of ensemble sampling will be based on a *master theorem*, presented in Section 3.1 and proven in Appendix B. The master theorem provides a method for obtaining regret bound for Thompson-sampling-like randomised algorithms. Thereafter, in Section 3.2, we apply the said master theorem to linear ensemble sampling, using some intermediate results proven in Sections 3.3 and 3.4, and some routine calculations that have been deferred to the appendix. Additionally, in Appendix D, we validate our master theorem by demonstrating that it recovers a previous result for a more classical Thompson-sampling-type algorithm.

### 3.1 Master Theorem: a regret bound for optimistic randomised algorithms

Our analysis of ensemble sampling will rely on the revered principle of *optimism*. To make this precise, consider a fixed instance parameter $\theta^\star \in \mathbb{R}^d$. Writing $J(\theta) = \max_{x \in \mathcal{X}} \langle x, \theta \rangle$, we call

$$\Theta^{\mathrm{OPT}} = \{\theta \in \mathbb{R}^d \colon J(\theta) \geq J(\theta^\star)\}$$

the set of parameters *optimistic* for $\theta^\star$. We now present a 'master theorem' that bounds the regret of any algorithm that chooses actions based on a randomly chosen parameter in terms of the probabilities that, given the past, the algorithm samples a parameter that falls in the intersection of ellipsoidal confidence sets and the optimistic region $\Theta^{\mathrm{OPT}}$. This result generalises a similar theorem stated for Thompson sampling by Abeille and Lazaric (2017), extending it to allow for finitely supported perturbations (critical for the proof of Theorem 1) and for a finer control over the way dependencies between time-steps are handled.

**Theorem 2** (Master regret bound). *Fix $T \in \mathbb{N}^+ \cup \{+\infty\}$, $\delta \in (0, 1]$, $\lambda \geq 1$, and let $(V_t, \hat{\theta}_t)_t$ be defined per equation* (1). *Let Assumption 1 hold, recalling the filtrations $(\mathcal{F}_t)_t$ and $(\mathcal{A}_t)_t$ defined therein, and let $(\mathcal{A}'_t)_t$ be any filtration satisfying*

$$\mathcal{A}_{t-1} \subset \mathcal{A}'_{t-1} \subset \mathcal{A}_t, \quad \forall t \in \mathbb{N}^+.$$

*Let $(b_t)_{t \in \mathbb{N}}$ be a sequence of $\sigma(\mathcal{F}_t \cup \mathcal{A}'_t)_t$-adapted nonnegative random variables, and define the ellipsoids*

$$\Theta_t = \hat{\theta}_t + b_t V_t^{-1/2} \mathbb{B}_2^d, \quad t \in \mathbb{N}.$$

Let $(\theta_t)_t$ be a $(\sigma(\mathcal{F}_t \cup \mathcal{A}_t))_t$-adapted $\mathbb{R}^d$-valued sequence and suppose that
$$X_t \in \arg\max_{x \in \mathcal{X}} \langle x, \theta_t \rangle, \quad \forall t \in [T].$$
Suppose further that there exist events $\mathcal{E}_T$ and $\mathcal{E}_T^\star$ satisfying
$$\mathcal{E}_T \subset \cap_{t=1}^T \{\theta_t \in \Theta_{t-1}\}, \quad \mathcal{E}_T^\star \subset \cap_{t=1}^T \{\theta^\star \in \Theta_{t-1}\} \quad and \quad \mathbb{P}(\mathcal{E}_T) \wedge \mathbb{P}(\mathcal{E}_T^\star) \geq 1 - \delta. \quad (2)$$
Then, writing
$$p_{t-1} = \mathbb{P}(\theta_t \in \Theta^{\mathrm{OPT}} \cap \Theta_{t-1} \mid \sigma(\mathcal{F}_{t-1} \cup \mathcal{A}'_{t-1})), \quad t \in \mathbb{N}^+,$$
we have that on a subset of $\mathcal{E}_T \cap \mathcal{E}_T^\star$ of probability at least $1 - 3\delta$, for all $\tau \in [T]$,
$$R(\tau) \leq 2 \max_{i \in [\tau]} \frac{b_{i-1}}{p_{i-1}} \left( 2\sqrt{2d\tau \log\left(1 + \frac{\tau}{d\lambda}\right)} + \sqrt{2(4\tau/\lambda + 1)\log\left(\frac{\sqrt{4\tau/\lambda + 1}}{\delta}\right)} \right).$$

We defer the proof of Theorem 2 to Appendix B.

**Remark 9.** *To apply the theorem, we need to show that our algorithm generates $(\theta_t)_t$ such that the probability of each $\theta_t$ landing in $\Theta^{\mathrm{OPT}} \cap \Theta_{t-1}$, conditional on $\sigma(\mathcal{F}_{t-1} \cup \mathcal{A}'_{t-1})$, is bounded away from zero for all $t \in [T]$. We have two degrees of freedom in our analysis:*

1. *We can choose $(b_t)_t$, the widths of the ellipsoids $(\Theta_t)_t$. Larger widths may make it easier to bound $(p_t)_t$ away from zero, but at a linear cost in the regret bound.*

2. *We can choose $\mathcal{A}'_{t-1}$, the 'point' between observing $Y_{t-1}$ and selecting $\theta_t$ with respect to which we consider the aforementioned conditional probabilities defined.*

**Remark 10.** *The introduction of $\mathcal{A}'_{t-1}$ serves to model the case where $\theta_t$ is sampled from a distribution $P_t$ that itself depends on $X_1, Y_1, \ldots, X_{t-1}, Y_{t-1}$ in a random manner. The $\sigma$-algebra $\mathcal{A}'_{t-1}$ is then such that $P_t$ is $\sigma(\mathcal{F}_{t-1} \cup \mathcal{A}'_{t-1})$-measurable.*

The upcoming two lemmas will be helpful our applications of the Master Theorem; both are stated in terms of the random functions $\psi_t \colon \mathbb{R}^d \to \mathbb{R}^d$ given by
$$\psi_t^\delta(u) = \hat{\theta}_t + \beta_t^\delta V_t^{-1/2} u, \quad t \in \mathbb{N}.$$
The first is the classic concentration result of Abbasi-Yadkori et al. (2011), and the second, Proposition 5 of Abeille and Lazaric (2017).

**Lemma 3.** *Under Assumption 1, for any $\delta \in (0, 1]$, $\mathbb{P}(\forall t \in \mathbb{N}, \; \theta^\star \in \psi_t^\delta(\mathbb{B}_2^d)) \geq 1 - \delta$.*

**Lemma 4.** *If Assumption 1 holds and $\theta^\star \in \psi_t^\delta(\mathbb{B}_2^d)$, then for any measure $Q$ over $\mathbb{R}^d$ and $a > 0$,*
$$Q(\Theta^{\mathrm{OPT}} \cap \psi_t^\delta(a\mathbb{B}_2^d)) \geq \inf_{u \in \mathbb{S}_2^{d-1}} Q(\psi_t^\delta(H_u \cap a\mathbb{B}_2^d)),$$
*where we recall that $H_u$ denotes the closed halfspace $\{v \in \mathbb{R}^d \colon \langle v, u \rangle \geq 1\}$.*

We provide a short, clean proof of Lemma 4 in Appendix C.

### 3.2 Proof of Theorem 1: regret bound for linear ensemble sampling

Let $\Gamma_0, \Gamma_1, \ldots$ be the sequence of random matrices in $\mathbb{R}^{d \times m}$ with the $j$th column given by
$$\Gamma_t^j = V_t^{-1/2} S_t^j, \quad \text{such that} \quad \tilde{\theta}_{t-1}^j = V_{t-1}^{-1/2} \Gamma_{t-1}^j.$$
Recall that $S_t^j = S_0^j + \sum_{i=1}^t U_i^j X_i$ and observe that each $(S^j)_t$ is a vector-valued random walk with increments given by $(U_t X_t)_t$, where $(X_t)_t$ is a serially correlated vector-valued sequence and $(U_t^j)_t$ is a sequence of independent $\mathcal{U}([-1, 1])$ variables, and $(\Gamma_t^j)_t$ is a normalised version of $(S_t^j)_t$. Consider the extremal singular values of $\Gamma_t$,
$$s_d(\Gamma_t) = \min_{u \in \mathbb{S}_2^{d-1}} \|\Gamma_t^\mathsf{T} u\|_2 = \min_{u \in \mathbb{S}_2^{d-1}} \left( \sum_{j=1}^m \langle \Gamma_t^j, u \rangle^2 \right)^{\frac{1}{2}} \quad \text{and} \quad s_1(\Gamma_t) = \|\Gamma_t^\mathsf{T}\| = \max_{u \in \mathbb{S}_2^{d-1}} \|\Gamma_t^\mathsf{T} u\|_2.$$

The lower of these may be interpreted as capturing how well the columns $\Gamma_t^1, \ldots \Gamma_t^m$ cover all directions $u \in \mathbb{S}_2^{d-1}$, and the upper, their maximal deviations from zero. The following theorem, proven in Section 3.3, shows that, for a sufficiently large $m$, with high probability, for all $t \in [T]$, all the singular values of $\Gamma_{t-1}$ are on the order of $\sqrt{m}$.

**Theorem 5.** *For $\lambda \geq 5$ and $m \geq 400 \log(3 + T) \vee 10d$, the event*

$$\mathcal{E}_T = \{\forall t \in [T], \ \sqrt{m}/7 \leq s_d(\Gamma_{t-1}) \leq s_1(\Gamma_{t-1}) \leq 10\sqrt{m}/7\}$$

*satisfies $\mathbb{P}(\mathcal{E}_T) \geq 1 - NTe^{-\frac{m}{400}}$, where $N = (134\sqrt{1 + T/\lambda})^d$.*

*Proof of Theorem 1.* We will use the master theorem, Theorem 2. We let the filtrations $(\mathcal{A}_t)_t$ and $(\mathcal{A}'_t)_t$ needed in this theorem be defined recursively via

$$\mathcal{A}'_0 = \sigma(S_0^1, \ldots, S_0^m), \ \ \mathcal{A}_t = \sigma(\mathcal{A}'_{t-1} \cup \sigma(J_t, \xi_t)) \ \text{ and } \ \mathcal{A}'_t = \sigma(\mathcal{A}_t \cup \sigma(U_t^1, \ldots, U_t^m)), \ t \in \mathbb{N}^+.$$

With this, the conditions concerning these filtrations hold.

We let $b_t = a\beta_t^\delta$ for $a = 10\sqrt{m}$, for each $t \in \mathbb{N}$, which is $\sigma(\mathcal{F}_t \cup \mathcal{A}'_t)_t$-adapted, as required. For the two required events, we take the event $\mathcal{E}_T$ from Theorem 5 and the event $\mathcal{E}_T^\star = \cap_{t \in \mathbb{N}}\{\theta_\star \in \psi_t(\mathbb{B}_2^d)\}$. To see that these satisfy the requirements given in equation (2), observe the following:

**Conditions on $\mathcal{E}_T$** By our assumptions on $m$ and $\lambda$, the conditions for Theorem 5 are satisfied. Hence, $\mathbb{P}(\mathcal{E}_T) \geq 1 - \delta$. We now show that on $\mathcal{E}_T$, for all $t \in [T]$, $\theta_t \in \Theta_{t-1}$. Note this is equivalent to the statement that on $\mathcal{E}_T$, for all $t \in [T]$, $\|\xi_t\Gamma_{t-1}^{J_t}\|_2 \leq b_{t-1}/r_{t-1} = 10\sqrt{m}/7$. And indeed, on $\mathcal{E}_T$,

$$\|\xi_t\Gamma_{t-1}^{J_t}\|_2 \leq \max_j \|\Gamma_{t-1}^j\|_2 \leq s_1(\Gamma_{t-1}) \leq 10\sqrt{m}/7, \quad \forall t \in [T]. \tag{3}$$

**Conditions on $\mathcal{E}_T^\star$** Since $a \geq 1$, $\psi_t(\mathbb{B}_2^d) \subset \psi_t(a\mathbb{B}_2^d) = \Theta_t$, and thus, by Lemma 3, $\theta_\star \in \Theta_t$ for all $t \in \mathbb{N}$ jointly with probability at least $1 - \delta$.

It remains to lower-bound the sequence $(p_t)_t$. For this, define $\mathbb{P}'_{t-1}(\cdot) = \mathbb{P}(\cdot \mid \sigma(\mathcal{F}_{t-1} \cup \mathcal{A}'_{t-1}))$. Fixing $t \in \mathbb{N}^+$, writing $Q(A) = \mathbb{P}'_{t-1}(\theta_t \in A)$, we have that on $\mathcal{E}_T^\star$,

$$
\begin{aligned}
p_{t-1} &= Q(\Theta^{\text{OPT}} \cap \Theta_{t-1}) \\
&\geq \inf_{u \in \mathbb{S}_2^{d-1}} Q(\psi_{t-1}^\delta(H_u \cap a\mathbb{B}_2^d)) && \text{(Lemma 4)} \\
&= \inf_{u \in \mathbb{S}_2^{d-1}} \mathbb{P}'_{t-1}(\psi_{t-1}^\delta(7\xi_t\Gamma_{t-1}^{J_t}) \in \psi_{t-1}^\delta(H_u \cap a\mathbb{B}_2^d)) && \text{(definition of } \theta_t, Q) \\
&= \inf_{u \in \mathbb{S}_2^{d-1}} \mathbb{P}'_{t-1}(7\xi_t\Gamma_{t-1}^{J_t} \in H_u \cap a\mathbb{B}_2^d) && (\psi_{t-1}^\delta \text{ is a bijection}) \\
&= \inf_{u \in \mathbb{S}_2^{d-1}} \frac{1}{2m} \sum_{(s,j) \in \{\pm 1\} \times [m]} \mathbf{1}[7s\Gamma_{t-1}^j \in H_u \cap a\mathbb{B}_2^d],
\end{aligned}
$$

where the last equality used the definitions of $\mathbb{P}'_{t-1}$, $\mathcal{F}_{t-1}$ and $\mathcal{A}'_{t-1}$.

We now show that on $\mathcal{E}_T$, regardless of the choice of $u$, for at least one $(s, j) \in \{\pm 1\} \times [m]$, $7s\Gamma_{t-1}^j \in H_u \cap a\mathbb{B}_2^d$. Assume thus that $\mathcal{E}_T$ holds. First observe that for any $(s, j)$, by equation (3), $7s\Gamma_{t-1}^j \in a\mathbb{B}_2^d$. Hence, it remains to show that for any $u$, for some $(s, j)$, $7s\Gamma_{t-1}^j \in H_u$; this holds because for any $u$,

$$1 \leq \frac{s_d^2(7\Gamma_{t-1})}{m} \leq \frac{1}{m}\sum_{j=1}^m \langle 7\Gamma_{t-1}^j, u\rangle^2 \leq \max_j \langle 7\Gamma_{t-1}^j, u\rangle^2. \tag{4}$$

Hence, on $\mathcal{E}_T \cap \mathcal{E}_T^\star$, $p_{t-1} \geq 1/(2m)$ and thus $b_{t-1}/p_{t-1} \leq 20m^{3/2}\beta_{t-1}^\delta$. Inserting this bound into the regret bound of Theorem 2 establishes the result. $\qquad \square$

**Remark 11** (On symmetrisation). *If the algorithm were run without symmetrisation, following the steps of the proof we see that we need to show that on $\mathcal{E}_T$, regardless the choice of $u$, for at least one of $j \in [m]$, $7\Gamma_{t-1}^j \in H_u$. In the presence of symmetrisation, the equivalent statement is that regardless of the choice of $u$, for at least one 'particle' $j$, we have either $7\Gamma_{t-1}^j \in H_u$ or $7\Gamma_{t-1}^j \in H_{-u}$. Through equation (4), the latter reduces to studying quadratic forms, which lead to convenient algebra.*

**Remark 12** (Can we improve our bound?)**.** *For any $u \in \mathbb{S}_2^{d-1}$, we lower bound the maximum $\max_j \langle 7\Gamma_{t-1}^j, u \rangle^2$ by the average $\frac{1}{m} \sum_{j=1}^m \langle 7\Gamma_{t-1}^j, u \rangle^2$, which we show exceeds 1, thus establishing the existence of at least one element $\pm 7\Gamma_{t-1}^j$ in $H_u$ (out of $2m$). This gives a $1/(2m)$ lower bound for $p_{t-1}$. However, lower bounding a maximum by an average might be wasteful. If, instead, we managed to lower bound the median (or any other fixed-proportion order statistic) of $\langle 7\Gamma_{t-1}^1, u \rangle^2, \ldots, \langle 7\Gamma_{t-1}^m, u \rangle^2$, we would be able to conclude that $p_t$ is lower bounded by a constant (since a constant proportion of 'particles' are then contained in $H_{\pm u}$), and thus conclude that linear ensemble sampling performs similarly to Thompson sampling (see Appendix D for an analysis of the latter). However, order statistics are considerably more technical to work with than averages, especially in a non-i.d.d. setting, and we do not currently know if such a result is attainable (for order statistics, see Gordon et al. (2012) and Litvak and Tikhomirov (2018) and the references within).*

**Remark 13** (Upper versus lower bound)**.** *When it comes to singular values, the difficulty is usually in the lower bounds. This is so for our problem as well. Consider our use of the upper bound in the middle inequality of equation* (3): *all we really needed there is that $\|\Gamma_{t-1}^j\|_2$ is upper bounded for all $j \in [m]$ and $t \in [T]$. But, for any fixed $j \in [m]$, we can apply the standard method-of-mixtures of bound of Abbasi-Yadkori et al. (2011) to obtain that $\|\Gamma_{t-1}^j\|_2 = O(\sqrt{d \log T})$ for all $t \in [T]$. An application of the union bound over the $j \in [m]$ then recovers what we need, up to logarithmic terms. We note that the $\sqrt{d}$ factor is strictly necessary, per Lattimore (2023).*

### 3.3  Setting up to prove Theorem 5: bound on singular values

Theorem 5 for $t = 0$ follows by standard methods (see Appendix E for proof):

**Lemma 6.** *Whenever $m \geq 10d$,*

$$\mathbb{P}\left( \sqrt{m/2} \leq s_d(\Gamma_0) \leq s_1(\Gamma_0) \leq \sqrt{3m/2} \right) \geq 1 - e^{-m/24}.$$

To extend the result to $t > 0$, we will consider the processes $(R_t^j(u))_t$ and $(R_t(u))_t$ defined for $u \in \mathbb{R}^d$, $u \neq 0$ by

$$R_t^j(u) = \frac{\langle u, S_t^j \rangle^2}{\|u\|_{V_t}^2} \quad \text{and} \quad R_t(u) = \frac{1}{m} \sum_j R_t^j(u).$$

Note that for $v = V_t^{1/2} u \neq 0$ one has $R_t^j(u) = \langle v, \Gamma_t^j \rangle^2 / \|v\|^2$. Since $V_t$ is positive-definite (and hence a bijection when viewed as a linear map) we observe the following relations:

**Claim 7.** *For all $t \geq 0$, $j \in [m]$,*

$$\sup_{u \neq 0} R_t^j(u) = \sup_{v \neq 0} \frac{\langle v, \Gamma_t^j \rangle^2}{\|v\|_2^2} = \|\Gamma_t^j\|_2^2 \quad \text{and} \quad \sup_{u \neq 0} R_t(u) = \sup_{v \neq 0} \frac{\|\Gamma_t^\mathsf{T} v\|_2^2}{m\|v\|_2^2} = \frac{s_1^2(\Gamma_t)}{m},$$

*and likewise $\inf_{u \neq 0} R_t(u) = s_d^2(\Gamma_t)/m$.*

In the upcoming section (Section 3.4) we establish the following bound on $R_t(u)$ for a fixed $u \in \mathbb{S}_2^{d-1}$.

**Lemma 8.** *Fix $u \in \mathbb{S}_2^{d-1}$ and $\lambda \geq 5$. Suppose that $m \geq 400 \log(3 + 2T)$. Then, there exists an event $\mathcal{E}_T'$ with $\mathbb{P}(\mathcal{E}_T') \geq 1 - Te^{-\frac{m}{400}}$ such that on $\mathcal{E}_T' \cap \{\frac{1}{2} \leq R_0(u) \leq \frac{3}{2}\}$,*

$$\frac{9}{100} \leq R_t(u) \leq \frac{5}{3}, \quad \forall t \in [T].$$

Our proof of Theorem 5 employs the above pointwise bound together with a covering argument over $\mathbb{S}_2^{d-1}$. For that, we need the following Lipschitzness result, proven in Appendix F (by simple algebra), and a standard bound on the size of $\epsilon$-nets of $\mathbb{S}_2^{d-1}$ (Lemma 4.10 in Pisier, 1999).

**Lemma 9.** *On $\mathbb{S}_2^{d-1}$, $R_t$ is $L$-Lipschitz with $L \leq 4\|\Gamma_t\|^2 \|V_t^{1/2}\|/(m\sqrt{\lambda})$.*

**Lemma 10.** *For all $\epsilon$ in $(0, 1]$, there exists an $\epsilon$-net $\mathcal{N}$ of $\mathbb{S}_2^{d-1}$ with $|\mathcal{N}| \leq \left(1 + \frac{2}{\epsilon}\right)^d$.*

*Proof of Theorem 5.* Let $\mathcal{N}_\epsilon$ be a minimal $\epsilon$-net of $\mathbb{S}_2^{d-1}$ and define the event

$$\mathcal{E}_\epsilon = \left\{ \forall v \in \mathcal{N}_\epsilon, \ \forall t \in [T], \ \frac{9}{100} \leq R_t(v) \leq \frac{5}{3} \right\}.$$

We will now confirm that, for a suitable choice of $\epsilon$, $\mathcal{E}_\epsilon$ is a subset of the event in Theorem 5, and that $\mathbb{P}(\mathcal{E}_\epsilon) \geq 1 - NTe^{-\frac{m}{400}}$, which establishes the theorem.

Let $u \in \mathbb{S}_2^{d-1}$ and $v \in \mathcal{N}_\epsilon$ be such that $\|u - v\|_2 \leq \epsilon$. From Lemma 9, and choosing $\epsilon = \sqrt{\lambda}/(132\sqrt{T + \lambda})$, and noting that $\|V_t^{1/2}\| \leq \sqrt{T + \lambda}$, we get

$$|R_t(u) - R_t(v)| \leq \frac{4\|\Gamma_t\|^2\|V_t^{1/2}\|}{m\sqrt{\lambda}}\epsilon \leq \frac{\|\Gamma_t\|^2}{33m}.$$

Then on $\mathcal{E}_\epsilon$, for our choice of $\epsilon$,

$$\|\Gamma_t\|^2 = m\sup_{u \neq 0} R_t(u) \leq m\sup_{v \in \mathcal{N}_\epsilon} R_t(v) + \frac{\|\Gamma_t\|^2}{33} \leq \frac{5}{3}m + \frac{\|\Gamma_t\|^2}{33},$$

Solving for $\|\Gamma_t\|^2$, we have that $\|\Gamma_t\|^2 = s_1^2(\Gamma_t) \leq \frac{165}{96}m$. The same argument also gives that, on $\mathcal{E}_\epsilon$,

$$s_d^2(\Gamma_t) \geq m\inf_{v \in \mathcal{N}_\epsilon} R_t(v) - \frac{\|\Gamma_t\|^2}{33} \geq \frac{9}{100}m - \frac{\|\Gamma_t\|^2}{33} \geq \frac{3}{100}m.$$

Loosening these bounds slightly, we have that on $\mathcal{E}_\epsilon$, $\sqrt{m}/7 \leq s_d(\Gamma_t) \leq s_1(\Gamma_t) \leq 10\sqrt{m}/7$.

The probability that $\mathcal{E}_\epsilon$ occurs is at least the probability that the event of Lemma 6 occurs and that the event of Lemma 8 occurs for each $u \in \mathcal{N}_\epsilon$, noting that the former ensures $\frac{1}{2} \leq R_0(u) \leq \frac{3}{2}$ for all $u \in \mathcal{N}_\epsilon$. Taking a union bound over these events, we have that

$$\mathbb{P}(\mathcal{E}_\epsilon) \geq 1 - e^{-\frac{m}{24}} - T|\mathcal{N}_\epsilon|e^{-\frac{m}{400}} \geq 1 - T(|\mathcal{N}_\epsilon| + 1)e^{-\frac{m}{400}}.$$

We conclude by noting that, by Lemma 10, for our choice of $\epsilon$, $N \geq |\mathcal{N}_\epsilon| + 1$. $\qquad\square$

### 3.4 Proof of Lemma 8

Since we now consider a fixed $u \in \mathbb{S}_2^{d-1}$, we will write $R_t^j := R_t^j(u)$ and $R_t := R_t(u)$. Let $(\mathcal{A}_t'')_t$ be the filtration given by $\mathcal{A}_t'' = \sigma(\mathcal{A}_t \cup \sigma(\xi_{t+1}, J_{t+1}))$ for each $t \in \mathbb{N}$, and let $\mathbb{E}_t''$ denote the $\sigma(\mathcal{F}_t \cup \mathcal{A}_t'' \cup \sigma(X_{t+1}))$-conditional expectation, which will be used to integrate out the random targets $U_{t+1}^1, \ldots, U_{t+1}^m$.

With that, we define

$$D_t = \mathbb{E}_t''R_{t+1} - R_t \quad \text{and} \quad W_{t+1} = R_{t+1} - \mathbb{E}_t''R_{t+1}$$

to be the drift and the noise of the process $(R_t)_{t \in \mathbb{N}}$, respectively. Also let

$$Q_t = \langle u, X_{t+1}\rangle^2/\|u\|_{V_{t+1}}^2$$

be the strength of the drift. These are related by the following two results:

**Claim 11.** $D_t = (\frac{2}{3} - R_t)Q_t$ for all $t \in \mathbb{N}$.

**Lemma 12.** For any $T \in \mathbb{N}^+$ and $m \geq 400\log(3 + T)$, there exists an event $\mathcal{E}$ with $\mathbb{P}(\mathcal{E}) \geq 1 - Te^{-\frac{m}{400}}$, such that on $\mathcal{E} \cap \{R_0 \leq 2\}$, for all $0 \leq \tau \leq t < T$,

$$\left|\sum_{i=\tau}^t W_{i+1}\right| \leq \frac{1}{10}\left(3 + \sum_{i=\tau}^t Q_iR_i\right).$$

The lemma is proven in Appendix G, and the claim in Appendix H. The constant $\frac{2}{3}$ above is just the almost sure value of $\mathbb{E}_t''[(U_{t+1}^j)^2]$; see also Remark 14 for a discussion of the significance of the $(U_{t+1}^j)^2$ terms and how we bound these in the proof of said lemma.

*Proof of Lemma 8.* Fix $0 \leq \tau \leq t < T$. We can decompose $R_{t+1}$ as

$$R_{t+1} = R_{t+1} - \mathbb{E}_t''R_{t+1} + \mathbb{E}_t''R_{t+1} - R_t + R_t = W_{t+1} + D_t + R_t,$$

which unrolled back to $\tau$ and combined with Claim 11 gives us that

$$R_{t+1} = R_\tau + \sum_{i=\tau}^t\left(\frac{2}{3} - R_i\right)Q_i + \sum_{i=\tau}^t W_{i+1}. \tag{5}$$

Observe from the above that the process $R_0, R_1, \dots$ drifts towards $\frac{2}{3}$, with strength proportional the level of deviation, scaled by $Q_i$. We will now show that on the event of Lemma 12, whenever $R_t$ moves sufficiently far away from $\frac{2}{3}$, the drift will overwhelm the effect of the noises $(W_t)_t$. Assume henceforth that the aforementioned event holds.

*Lower bound.* We consider the excursions of $(R_t)_t$ where it goes and stays below $1/2$. Let $0 \leq \tau < s \leq T$ be endpoints of such a maximal excursion, in the sense that $R_\tau \geq 1/2$, $R_{\tau+1}, \dots, R_s < 1/2$ and if $s + 1 \leq T$ then $R_{s+1} \geq 1/2$. Our goal is to show that for any $\tau \leq t < s$, $R_{t+1} \geq 9/100$, which suffices to prove the lower bound. Fix $t \in [\tau, s)$. From equation (5) and since the event from Lemma 12 holds, defining $\alpha = 1/10$,

$$
\begin{aligned}
R_{t+1} &\geq R_\tau + \sum_{i=\tau}^{t} \left( \frac{2}{3} - R_i \right) Q_i - \alpha \left( 3 + \sum_{i=\tau}^{t} Q_i R_i \right) \\
&= (1 - (1+\alpha)Q_\tau)R_\tau + \tfrac{2}{3}Q_\tau - 3\alpha + \sum_{i=\tau+1}^{t} \left( \frac{2}{3} - (1+\alpha)R_i \right) Q_i \\
&\geq (1 - (1+\alpha)Q_\tau)R_\tau + \tfrac{2}{3}Q_\tau - 3\alpha + \sum_{i=\tau+1}^{t} \left( \frac{2}{3} - \frac{11}{20} \right) Q_i \\
&\hspace{6cm} (R_{\tau+1}, \dots, R_t < 1/2, \text{ def. of } \alpha) \\
&\geq (1 - (1+\alpha)Q_\tau)R_\tau - 3\alpha \hspace{2.7cm} (Q_i \geq 0, \tfrac{2}{3} - \tfrac{11}{20} > 0) \\
&\geq (1 - \tfrac{11}{50})\tfrac{1}{2} - \tfrac{3}{10} \hspace{2.5cm} (Q_i \leq 1/\lambda \leq 1/5, R_\tau \geq 1/2, \text{ def. of } \alpha) \\
&= \tfrac{9}{100} .
\end{aligned}
$$

*Upper bound.* The upper bound follows near-verbatim, taking $\tau$ with $R_\tau \leq \frac{3}{2} < R_{\tau+1}$. $\qquad\qquad \square$

**Remark 14.** *The proof of Lemma 8 was where we use that the targets $(U_t^j)$ are uniform—or, in particular, that they are bounded random variables—for each $W_{t+1}$ features $(U_t^j)^2$ terms, and might otherwise be only sub-exponential. Of course, in that case, we would simply use a truncation argument: pick some truncation level $a > 0$, set $W'_{t+1} = W_{t+1} \wedge a$ for each $t \in \mathbb{N}^+$ and work with the process given by the recursion $R'_{t+1} = W'_{t+1} + D_t + R'_t$. Then, $R_t \geq R'_t$ for all $t \in \mathbb{N}$, and the truncated noises $(W'_{t+1})_t$ are once again subgaussian, so our approach to lower bounding $R_t$ would also work for $R'_t$. We could then establish what we needed from the upper bound (that is, a bound for the quantity in equation (3)) as discussed in Remark 13, which does not require bounded targets.*

## 4 Discussion

We showed that linear ensemble sampling can work with relatively small ensembles, providing the first useful theoretical grounding for the method (see Remarks 6 and 7 for comparisons). We do, however, believe our result to be loose (see Remarks 2 and 12 for discussion); resolving this would make for an important step forward. Moreover, our algorithm uses a certain symmetrisation not used within the work of Lu and Van Roy (2017) (see Section 2.2 and Remark 11). While this symmetrisation comes with no particular downsides, we would nonetheless be curious to see whether there is a clean way of making the analysis go through without it. A natural further question is whether the idea of adding noise to the rewards is the right approach to the explore-vs-exploit dilemma. On this, first, in Janz et al. (2024) we showed that beyond the linear setting, it is losses rather than rewards that should be perturbed—with the two approaches being equivalent in the linear-Gaussian setting. Second, the very recent work of Cassel et al. (2024) shows that in the multi-armed bandit setting, a simple bootstrap-based method, which takes a max over the ensemble (as in Remark 5), yields instance dependent bounds. This raises the question of the trade-offs, if any, between randomising over the ensemble and taking a maximum, and between bootstrapping and using perturbations.

## Acknowledgements

We thank Alireza Bakhtiari for proofreading the manuscript. Csaba Szepesvári gratefully acknowledges funding from the Canada CIFAR AI Chairs Program, Amii and NSERC.

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

## A A reformulation of Algorithm 1 in the style of Lu and Van Roy (2017)

Below in Algorithm 2, we list a reformulation of Algorithm 1 written using updates in the style of Lu and Van Roy (2017), with one difference: the algorithm here accepts only a single time-stationary perturbation scale $r > 0$ (this is unavoidable with the Lu and Van Roy (2017) style of updates). Taking $r = 7\tilde{\beta}_T^\delta$, where $\tilde{\beta}_T^\delta$ is the deterministic upper bound on $\beta_0^\delta, \ldots, \beta_T^\delta$ given by

$$\tilde{\beta}_T^\delta = \sqrt{\lambda} + \sqrt{2\log(1/\delta) + d\log((d + T/\lambda)/d)} \, ,$$

allows for the same guarantee as given in Theorem 1, but with $\tilde{\beta}_T^\delta$ in place of $\beta_{\tau-1}^\delta$.

---

**Algorithm 2** Equivalent form of Algorithm 1 assuming time-stationary perturbation scale $r$

---

**Input** regularisation parameter $\lambda > 0$, half-ensemble size $m \in \mathbb{N}^+$, perturbation scale $r > 0$
Let $V_0 = \lambda I$, sample $(w_0^j) \sim \mathcal{U}(\sqrt{d}\mathbb{S}_2^{d-1})^{\otimes m}$ and let $w_0^{m+j} = -w_0^j$ for each $j \in [m]$
**for** $t \in \mathbb{N}^+$ **do**
  Sample $J_t \sim \mathcal{U}(\{1, \ldots, 2m\})$
  Compute an $X_t \in \arg\max_{x \in \mathcal{X}} \langle x, w_{t-1}^{J_t} \rangle$, play action $X_t$ and receive reward $Y_t$
  Sample $(U_t^j)_{j \in [m]} \sim \mathcal{U}([-1, 1])^{\otimes m}$ and let $U_t^{m+j} = -U_t^j$ for each $j \in [m]$
  Let $V_t = V_{t-1} + X_t X_t^\mathsf{T}$, and let $w_t^j = V_t^{-1}(V_{t-1} w_{t-1}^j + X_t(Y_t + rU_t^j))$ for each $j \in [2m]$

---

To confirm that Algorithm 2 is equivalent to Algorithm 1 if the sequence $r_0, r_1, \ldots$ is set to the common value $r$, it suffices to compare the list of parameters $w_{t-1}^1, \ldots, w_{t-1}^{2m}$ used here with the list

$$\hat{\theta}_{t-1} \pm r\tilde{\theta}_{t-1}^1, \ldots, \hat{\theta}_{t-1} \pm r\tilde{\theta}_{t-1}^1 \, ,$$

at each step of the algorithm. We leave this as a simple algebraic exercise for the reader.

## B Proof of Theorem 2: master regret bound

We need the following concentration inequality, a simple consequence of Exercise 20.8 in Lattimore and Szepesvári (2020)[2] and Hoeffding's lemma (Lemma 2.2, Boucheron et al., 2013), and the elliptical potential lemma (Lemma 19.4 in Lattimore and Szepesvári, 2020).

**Lemma 13.** *Fix $0 < \delta \leq 1$. Let $(\xi_t)_{t \in \mathbb{N}^+}$ be a real-valued martingale difference sequence satisfying $|\xi_t| \leq c$ almost surely for each $t \in \mathbb{N}^+$ and some $c > 0$. Then,*

$$\mathbb{P}\left(\exists \tau: \left(\sum_{t=1}^\tau \xi_t\right)^2 \geq 2(c^2\tau + 1)\log\left(\frac{\sqrt{c^2\tau + 1}}{\delta}\right)\right) \leq \delta.$$

**Lemma 14** (Elliptical potential lemma). *Let $(x_t)_{t \in \mathbb{N}^+}$ be a sequence of vectors in $\mathbb{B}_2^d$, let $V_0 = \lambda I$ for some $\lambda \geq 1$ and $V_t = V_0 + \sum_{i=1}^t x_i x_i^\mathsf{T}$ for each $t \in \mathbb{N}^+$. Then, for all $\tau \in \mathbb{N}^+$,*

$$\sum_{t=1}^\tau \|x_t\|_{V_{t-1}^{-1}}^2 \leq 2d\log\left(1 + \frac{\tau}{\lambda d}\right).$$

Recall that $J(\theta) = \max_{x \in \mathcal{X}} \langle x, \theta \rangle$ is the support function of $\mathcal{X}$, and observe the following:

**Claim 15.** *For any $t \in \mathbb{N}^+$, $X_t$ is a subgradient of $J$ at $\theta_t$.*

*Proof.* Fix $t \in \mathbb{N}^+$. For any $\theta \in \mathbb{R}^d$,

$$J(\theta_t) + \langle X_t, \theta - \theta_t \rangle = \langle X_t, \theta_t \rangle + \langle X_t, \theta - \theta_t \rangle = \langle X_t, \theta \rangle \leq \max_{x \in \mathcal{X}}\langle x, \theta \rangle = J(\theta), \tag{6}$$

which is the inequality that defines a subgradient. $\qquad\square$

---

[2]The statement of this result within said exercise contains a typographical error, in that the $c^2$ appears outside the brackets, rather than multiplying just the $\tau$. The version here is the correct result.

*Proof of Theorem 2.* Throughout this proof, we work on the intersection $\mathcal{E}_T \cap \mathcal{E}_T^\star$, and therefore in particular use that for any $t \in [T]$, $\theta_t, \theta^\star \in \Theta_t$. We let $\gamma_t = 2b_t$, and observe that this is the $V_t$-weighted Euclidean norm width of $\Theta_t$, in the sense that

$$\gamma_t = \max\{\|\theta - \theta'\|_{V_t} : \theta, \theta' \in \Theta_t\}. \tag{7}$$

We will also use the shorthands $\mathbb{F}'_t = \sigma(\mathcal{F}_t \cup \mathcal{A}'_t)$, $\mathbb{P}'_t = \mathbb{P}(\cdot \mid \mathbb{F}'_t)$ and $\mathbb{E}'_t = \mathbb{E}[\cdot \mid \mathbb{F}'_t]$.

The proof is based on decomposing the regret into two parts, which we then control separately:

$$R(\tau) = \sum_{t=1}^{\tau} (J(\theta^\star) - J(\theta_t)) + \sum_{t=1}^{\tau} (J(\theta_t) - \langle X_t, \theta^\star \rangle). \tag{8}$$

Fix $t \in [\tau]$ and consider the second term, $J(\theta_t) - \langle X_t, \theta^\star \rangle$. We have that

$$J(\theta_t) - \langle X_t, \theta^\star \rangle = \langle X_t, \theta_t - \theta^\star \rangle \le \|X_t\|_{V_{t-1}^{-1}} \|\theta_t - \theta^\star\|_{V_{t-1}} \le \gamma_{t-1}\|X_t\|_{V_{t-1}^{-1}}, \tag{9}$$

where the first inequality is by Cauchy-Schwartz, the second uses that $\theta_t, \theta^\star \in \Theta_{t-1}$ and then equation (7).

Now consider the first term, $J(\theta^\star) - J(\theta_t)$. Let $\theta^-$ be a minimiser $J$ over $\Theta_{t-1}$ (which is well-defined, since $J$ is continuous and $\Theta_{t-1}$ closed) and let $\theta^+$ be an element of $\Theta^{\mathrm{OPT}} \cap \Theta_{t-1}$ (which is non-empty, since it contains at least $\theta^\star$). Then, since $\theta^\star, \theta_t \in \Theta_{t-1}$, we have the bound

$$J(\theta^\star) - J(\theta_t) \le J(\theta^\star) - J(\theta^-) \le J(\theta^+) - J(\theta^-).$$

It follows that for any probability measure $Q$ over $\Theta^{\mathrm{OPT}} \cap \Theta_{t-1}$,

$$J(\theta^\star) - J(\theta_t) \le \int J(\theta^+) - J(\theta^-) \, \mathrm{d}Q(\theta^+).$$

Let $\Theta_{t-1}^{\mathrm{OPT}} = \Theta^{\mathrm{OPT}} \cap \Theta_{t-1}$, and take $Q = Q_{t-1}$ in the integral above defined by

$$Q_{t-1} = \begin{cases} \mathbb{P}'_{t-1}(\theta_t \in \cdot \cap \Theta_{t-1}^{\mathrm{OPT}})/p_{t-1}, & p_{t-1} > 0; \\ \text{any arbitrary probability measure}, & \text{otherwise}, \end{cases}$$

which yields the bound

$$J(\theta^\star) - J(\theta_t) \le \frac{1}{p_{t-1}} \mathbb{E}'_{t-1}[(J(\theta_t) - J(\theta^-))\mathbf{1}[\theta_t \in \Theta_{t-1}^{\mathrm{OPT}}]],$$

where for $p_{t-1} = 0$ we take the upper bound to be positive infinity. Observing that $X_t$ is a subgradient of $J$ at $\theta_t$ (Claim 15 and equation (6)) and applying Cauchy-Schwartz, we have that

$$J(\theta_t) - J(\theta^-) \le \langle X_t, \theta_t - \theta^- \rangle \le \|X_t\|_{V_{t-1}^{-1}} \|\theta^- - \theta_t\|_{V_{t-1}},$$

Moreover, since $\theta^- \in \Theta_{t-1}$, observing that norms are non-negative, that $\Theta_{t-1}^{\mathrm{OPT}} \subset \Theta_{t-1}$ and using equation (7),

$$\|\theta^- - \theta_t\|_{V_{t-1}}\mathbf{1}[\theta_t \in \Theta_{t-1}^{\mathrm{OPT}}] \le \|\theta^- - \theta_t\|_{V_{t-1}}\mathbf{1}[\theta_t \in \Theta_{t-1}] \le \gamma_{t-1},$$

and therefore, since $\gamma_{t-1}$ is $\mathbb{F}'_{t-1}$-measurable (by assumption), we have the bound

$$\frac{1}{p_{t-1}} \mathbb{E}'_{t-1}[(J(\theta_t) - J(\theta^-))\mathbf{1}[\theta_t \in \Theta_{t-1}^{\mathrm{OPT}}]] \le \frac{\gamma_{t-1}}{p_{t-1}} \mathbb{E}'_{t-1}[\|X_t\|_{V_{t-1}^{-1}}].$$

Chaining the above inequalities and writing these in terms of $\Delta_t = \mathbb{E}'_{t-1}[\|X_t\|_{V_{t-1}^{-1}}] - \|X_t\|_{V_{t-1}^{-1}}$, we have the bound

$$J(\theta^\star) - J(\theta_t) \le \frac{\gamma_{t-1}}{p_{t-1}} \mathbb{E}'_{t-1}[\|X_t\|_{V_{t-1}^{-1}}] = \frac{\gamma_{t-1}}{p_{t-1}} \left(\|X_t\|_{V_{t-1}^{-1}} + \Delta_t\right), \tag{10}$$

Combining equations (9) and (10) with the regret decomposition in equation (8),

$$R(\tau) \le \sum_{t=1}^{T} \left(\left(\gamma_{t-1} + \frac{\gamma_{t-1}}{p_{t-1}}\right)\|X_t\|_{V_{t-1}^{-1}} + \frac{\gamma_{t-1}}{p_{t-1}}\Delta_t\right) \le \max_{i \in [\tau]} \frac{\gamma_{i-1}}{p_{i-1}} \left(2\sum_{t=1}^{\tau} \|X_t\|_{V_{t-1}^{-1}} + \sum_{t=1}^{\tau} \Delta_t\right), \tag{11}$$

for any $\tau \in [T]$. For the first sum in that upper bound, by Cauchy-Schwartz and the elliptical potential lemma (Lemma 14, which can be applied because by assumption $\lambda \geq 1$), for any $\tau \in \mathbb{N}^+$,

$$\sum_{t=1}^{\tau} \|X_t\|_{V_{t-1}^{-1}} \leq \left(\tau \sum_{t=1}^{\tau} \|X_t\|_{V_{t-1}^{-1}}^2\right)^{\frac{1}{2}} \leq \sqrt{2\tau d \log\left(1 + \frac{\tau}{d\lambda}\right)}. \tag{12}$$

To deal with the second sum, observe that since for all $t \in \mathbb{N}^+$, $V_{t-1} \succeq \lambda I$ and $X_t \in \mathbb{B}_2^d$,

$$\|X_t\|_{V_{t-1}^{-1}}^2 = \langle X_t, V_{t-1}^{-1} X_t \rangle \leq \|X_t\|_2^2/\lambda \leq 1/\lambda \quad \text{and so} \quad |\Delta_t| \leq 2/\sqrt{\lambda} \quad \text{for all } t \in \mathbb{N}^+.$$

Moreover, observe that $(\Delta_t)_t$ is an $(\mathbb{F}'_t)_t$-adapted martingale difference sequence. We thus apply Lemma 13 with $c = 2/\sqrt{\lambda}$, to obtain the deviation probability bound

$$\mathbb{P}\left(\exists \tau \in \mathbb{N}^+ : \sum_{t=1}^{\tau} \Delta_t \geq \sqrt{2(4\tau/\lambda + 1) \log\left(\frac{\sqrt{4\tau/\lambda + 1}}{\delta}\right)}\right) \leq \delta. \tag{13}$$

The bounds on the two sums, equation (12) and equation (13), when inserted into equation (11) and combined with a union bound, yield the claimed result. $\qquad \square$

## C  Proof of Lemma 4: optimism for elliptical confidence sets

**Lemma 16.** *Let $F : \mathbb{R}^d \to \mathbb{R}$ be a convex function and let $u$ be its maximizer over the unit ball. Then, for any $v \in H_u \doteq \{v \in \mathbb{R}^d : \langle v, u \rangle \geq 1\}$, we have $F(v) \geq F(u)$.*

*Proof.* For any $v \in \mathbb{R}^d$ with $\langle v, u \rangle > 1$, the ray from $v$ to $u$ enters the interior of the unit ball. Hence, for any such $v$, there exists a $z \in \mathbb{B}_2^d$ and $\alpha \in (0, 1)$ such that $u = \alpha z + (1 - \alpha)v$. By convexity and maximality,

$$F(u) = F(\alpha z + (1 - \alpha)v) \leq \alpha F(z) + (1 - \alpha)F(v) \leq \alpha F(u) + (1 - \alpha)F(v).$$

Hence, $F(u) \leq F(v)$. Since any finite convex function on an open set is continuous, the result holds for any $v \in H_u$. $\qquad \square$

*Proof of Lemma 4.* Write $F = J \circ \psi_t^{\delta}$; since $J$ is convex and $\psi_t^{\delta}$ is affine, $F$ is convex. Let $u^+$ be the maximiser of $F$ over $\mathbb{B}_2^d$. Since $F$ is a convex function and $\mathbb{B}_2^d$ is a convex set, $u^+ \in \partial \mathbb{B}_2^d = \mathbb{S}_2^{d-1}$. By assumption, $\theta^\star \in \psi_t^{\delta}(\mathbb{B}_2^d)$, and so $J(\theta^\star) \leq F(u^+)$. By Lemma 16, $F(u^+) \leq F(u')$ for any $u' \in H_{u^+}$. Thus, $\psi_t^{\delta}(H_{u^+}) \subset \Theta^{\mathrm{OPT}}$, and so

$$\Theta^{\mathrm{OPT}} \cap \psi_t^{\delta}(a\mathbb{B}_2^d) \supset \psi_t^{\delta}(H_{u^+}) \cap \psi_t^{\delta}(a\mathbb{B}_2^d) \supset \psi_t^{\delta}(H_{u^+} \cap a\mathbb{B}_2^d).$$

Therefore, for any measure $Q$ on $\mathbb{R}^d$,

$$Q(\Theta^{\mathrm{OPT}} \cap \psi_t^{\delta}(a\mathbb{B}_2^d)) \geq Q(\psi_t^{\delta}(H_{u^+} \cap a\mathbb{B}_2^d)) \geq \inf_{u \in \mathbb{S}_2^{d-1}} Q(\psi_t^{\delta}(H_u \cap a\mathbb{B}_2^d)). \qquad \square$$

## D  Analysis of Thompson sampling via our master theorem

Algorithm 3 is a version of Thompson sampling we call *confident linear Thompson sampling*. It is extremely simple: at each step $t \in [T]$, it picks an action $X_t$ that is optimal according to an estimate $\theta_t$ sampled uniformly on $\psi_{t-1}^{\delta}(\sqrt{d}\mathbb{B}_2^d) = \Theta_{t-1}$, a $\sqrt{d}$-inflation of the ridge regression confidence set. It differs from usual linear Thompson sampling of Agrawal and Goyal (2013) through the use of a uniform sampling distribution.

**Remark 15.** *Our use of the uniform distribution to generate perturbed parameters is purely for the sake of a clean exposition, and for easy comparison with our linear ensemble sampling algorithm. Observe that the usual analysis for the Gaussian (or subgaussian) case begins by restricting to a high-probability event where every $\theta_t$ lands within some inflated version of the corresponding $\Theta_{t-1}$ (as in Agrawal and Goyal, 2013; Abeille and Lazaric, 2017).*

---
**Algorithm 3** Confident linear Thompson sampling
---
**for** $t \in \mathbb{N}^+$ **do**
  Sample $U_t \sim \mathcal{U}(\sqrt{d}\mathbb{B}_2^d)$ and compute $\theta_t = \psi_{t-1}^\delta(U_t)$
  Compute some $X_t \in \arg\max_{x \in \mathcal{X}} \langle x, \theta_t \rangle$, play action $X_t$ and receive reward $Y_t$
---

**Theorem 17.** *Let Assumption 1 hold. Fix $\delta \in (0,1]$ and let $\lambda \geq 1$. There exist some universal constant $C > 0$ such that with probability $1 - \delta$, a learner using Algorithm 3 incurs regret satisfying*

$$R(T) \leq C\beta_{\tau-1}^\delta \sqrt{d} \left( \sqrt{d\tau \log(1 + \tau/(\lambda d))} + \sqrt{(\tau/\lambda) \log(\tau/(\lambda\delta))} \right) \quad \text{for all} \quad \tau \in [T].$$

The above result recovers the same regret bound for confident linear Thompson sampling as that given by the analysis of Abeille and Lazaric (2017). The proof is as follows.

*Proof of Theorem 17.* Take $T = \infty$. Fix $b_t = \sqrt{d}$ for all $t \in \mathbb{N}$. As each $\theta_t$ is a uniform random variable on $\Theta_{t-1}$ for all $t \in \mathbb{N}$, and so event $\mathcal{E}_T$ holds almost surely. Moreover, as for ensemble sampling, we take $\mathcal{E}_T^\star$ to be the event from the standard concentration result for ridge regression, Lemma 3, observing as before that, $\psi_{t-1}^\delta(\mathbb{B}_2^d) \subset \psi_{t-1}^\delta(\sqrt{d}\mathbb{B}_2^d) = \psi_{t-1}^\delta(b_t\mathbb{B}_2^d)$.

We pick $\mathcal{A}'_{t-1} = \mathcal{A}_{t-1}$ for all $t \in \mathbb{N}^+$. Now, to lower bound each $p_{t-1}$, note that on $\mathcal{E}_T^\star$, by Lemma 4 applied with $Q(A) = \mathbb{P}(\theta_t \in A \mid \sigma(\mathcal{F}_{t-1} \cup \mathcal{A}'_{t-1})) =: \mathbb{P}'_{t-1}(\theta_t \in A)$,

$$p_{t-1} = \mathbb{P}'_{t-1}(\theta_t \in \Theta^{\mathrm{OPT}} \cap \Theta_{t-1}) \geq \inf_{u \in \mathbb{S}_2^{d-1}} \mathbb{P}'_{t-1}(\theta_t \in \psi_{t-1}^\delta(H_u \cap \sqrt{d}\mathbb{B}_2^d)).$$

And, since $\psi_{t-1}^\delta$ is a bijection and $U_t$ is uniform on $\sqrt{d}\mathbb{B}_2^d$, and thus rotationally invariant,

$$\inf_{u \in \mathbb{S}_2^{d-1}} \mathbb{P}'_{t-1}(\theta_t \in \psi_{t-1}^\delta(H_u \cap \sqrt{d}\mathbb{B}_2^d)) = \inf_{u \in \mathbb{S}_2^{d-1}} \mathbb{P}'_{t-1}(U_t \in H_u \cap \sqrt{d}\mathbb{B}_2^d) = \mathcal{U}(H_1 \cap \sqrt{d}\mathbb{B}_2^d).$$

As established in Appendix A of Abeille and Lazaric (2017), $\mathcal{U}(H_1 \cap \sqrt{d}\mathbb{B}_2^d) \geq 1/(16\sqrt{3\pi})$, independently of $d$. This means that, on $\mathcal{E}_T^\star$, $\frac{b_{t-1}}{p_{t-1}} \leq 16\sqrt{3\pi d}$ for all $t \in \mathbb{N}^+$; plugging this into the regret bound of Theorem 2 yields the claim. $\qquad\square$

# E  Proof of Lemma 6: singular values at initialisation

Lemma 6 follows by taking $y = \frac{81m}{2500}$ and $\epsilon = \frac{1}{20}$ in Theorem 18.

**Theorem 18.** *Let $U \in \mathbb{R}^{m \times d}$, $m \geq d$, be a random matrix with independent rows $U_1, \ldots, U_m$ distributed uniformly on $\sqrt{d}\mathbb{S}_2^{d-1}$. Then, for all $y > 0$, $\epsilon \in (0, 1/2)$, and with $c_\epsilon = 1/(1 - 2\epsilon)$,*

$$\mathbb{P}\{\sqrt{m} - c_\epsilon\sqrt{y} \leq s_d(U) \leq s_1(U) \leq \sqrt{m} + c_\epsilon\sqrt{y}\} \geq 1 - \exp\{-3y/8 + \log(2(1 + 2/\epsilon)^d)\}.$$

The proof of Theorem 18 follows the approach of Chapter 4 of Vershynin (2018), but makes the constants explicit. For these constants, we will need the following claim:

**Claim 19.** *Fix $x \in \mathbb{S}_2^{d-1}$, let $X \sim \mathcal{U}(\mathbb{S}_2^{d-1})$ and $X_x^2 = \langle X, x \rangle^2$. Then, $\mathbb{E}X_x^2 = 1/d$, and for some $\nu, c > 0$ satisfying $\nu \leq 2/d^2$ and $c \leq 4/d$ and all $0 < s < 1/c$,*

$$\mathbb{E}\exp(s\,|X_x^2 - \mathbb{E}X_x^2|) \leq \exp\left(\frac{s^2\nu/2}{1 - cs}\right).$$

*Proof.* It is known that $X_x^2 \sim \mathrm{Beta}(\frac{1}{2}, \frac{d-1}{2})$ (see, for example, Theorem 1.5 and the discussion thereafter in Fang, 1990), the expectation of which is $1/d$. We thus need only look up moment generating function bounds for beta random variables. Skorski (2023) derives such in their proof of their Theorem 1, and our result follows by substituting in the parameters of our beta distribution, and bounding the resulting $\nu$ and $c$ crudely. $\qquad\square$

We will need the next two results from Vershynin (2018), appearing as Lemma 4.1.5 and Exercise 4.4.3 respectively.

**Lemma 20** (Appropximate isometry). *For any matrix $A \in \mathbb{R}^{m \times d}$ and $\epsilon \geq 0$,*
$$\|A^\mathsf{T} A - I\| \leq \epsilon \vee \epsilon^2 \implies 1 - \epsilon \leq s_d(A) \leq s_1(A) \leq 1 + \epsilon.$$

**Lemma 21** (Quadratic form on a net). *For a symmetric matrix $A \in \mathbb{R}^{d \times d}$ and an $\epsilon$-net $\mathcal{N}$ of $\mathbb{S}_2^{d-1}$ with $\epsilon \in (0, 1/2)$,*
$$\|A\| \leq \frac{1}{1 - 2\epsilon} \sup_{x \in \mathcal{N}} |\langle Ax, x \rangle|.$$

*Proof of Theorem 18.* Fix $x \in \mathbb{S}_2^{d-1}$, and consider $Z_x^2 = \frac{1}{m}\|Ux\|_2^2 = \frac{d}{m} \sum_{j=1}^m \langle U_j/\sqrt{d}, x \rangle^2$. Observe that each $U_j/\sqrt{d} \sim \mathcal{U}(\mathbb{S}_2^{d-1})$; thus, by Claim 19 and since $U_1, \ldots, U_m$ are independent, adopting the notation of the claim, we have that, for all $0 < sd/m < 1/c$,
$$\mathbb{E} \exp(s\,|Z_x^2 - 1|) = \prod_{j=1}^m \mathbb{E} \exp\left(\frac{sd}{m}|X_x^2 - \mathbb{E}X_x^2|\right) \leq \exp\left(\frac{s^2 d^2 \nu/(2m)}{1 - csd/m}\right).$$

Thus, $|Z_x^2 - 1|$ is what Boucheron et al. (2013) would term sub-gamma with parameters $(d^2\nu/m,\ cd/m)$ on both tails. Applying the maximal-form of the there-stated Bernstein-type bound for sub-gamma random variables, a union bound over the two tails, and the bounds $\nu \leq 2/d^2$ and $c \leq 4/d$ from Claim 19, we have that, for all $y > 0$,
$$\mathbb{P}(|Z_x^2 - 1| \geq \sqrt{y/m} \vee y/m) \leq 2e^{-3y/8}.$$

Now let $\mathcal{N}_\epsilon$, $\epsilon \in (0, 1/2)$, be an $\epsilon$-net of $\mathbb{S}_2^{d-1}$. By Lemma 21,
$$\sup_{x \in \mathbb{S}_2^{d-1}} |Z_x^2 - 1| \leq \frac{1}{1 - 2\epsilon} \max_{x \in \mathcal{N}_\epsilon} |Z_x^2 - 1|.$$

Also, by our bound on nets from Lemma 10, $|\mathcal{N}_\epsilon| \leq (1 + 2/\epsilon)^d$. Thus, taking a union bound over $x \in \mathcal{N}_\epsilon$, we have that for any $y > 0$ and $\epsilon \in (0, 1/2)$, the probability that
$$\mathbb{P}\left(\sup_{x \in \mathbb{S}_2^{d-1}} |Z_x^2 - 1| \leq \frac{1}{1 - 2\epsilon}\left(\sqrt{y/m} \vee y/m\right)\right) \geq 1 - \exp\{-3y/8 + \log(2(1 + 2/\epsilon)^d)\}.$$

We conclude the proof by observing that $\sup_{x \in \mathbb{S}_2^{d-1}} |Z_x^2 - 1| = \|U^\mathsf{T} U/m - I\|$ and applying Lemma 20. $\qquad\square$

## F  Proof of Lemma 9: Lipschitzness result

*Proof of Lemma 9.* Fix $u \neq 0$ and let $z = V_t^{1/2} u$. Then,
$$R_t(u) = \frac{1}{m} \sum_{j=1}^m R_t^j(u) = \frac{1}{m} \sum_{j=1}^m \frac{\langle z, \Gamma_t^j \rangle^2}{\|z\|_2^2} = \frac{\|\Gamma_t^\mathsf{T} z\|_2^2}{m\|z\|_2^2}.$$

Now note that for all non-negative $a, b, A, B$ with $b \geq a$,
$$\left|\frac{A^2}{a^2} - \frac{B^2}{b^2}\right| = \left|\frac{A^2(b^2 - a^2) + (A^2 - B^2)a^2}{a^2 b^2}\right| \leq \frac{2A^2}{a^2}\frac{|b - a|}{b} + \frac{|A - B|(A + B)}{b^2}.$$

Let $u, v \in \mathbb{S}_2^{d-1}$, $z = V_t^{1/2} u$, $w = V_t^{1/2} v$. Let $\epsilon = \|u - v\|_2$. Let $A = \|\Gamma_t^\mathsf{T} z\|_2$, $B = \|\Gamma_t^\mathsf{T} w\|_2$, $a = \|z\|_2$, $b = \|w\|_2$. Assume without loss of generality that $b \geq a$. Since $v \in \mathbb{S}_2^{d-1}$ and $b \geq \sqrt{\lambda}$,
$$2\frac{A^2}{a^2}\frac{|b - a|}{b} \leq \frac{2\|\Gamma_t\|^2 \|z - w\|_2}{\sqrt{\lambda}} \leq 2\|\Gamma_t\|^2 \frac{\|V_t^{1/2}\|}{\sqrt{\lambda}} \epsilon,$$

and likewise
$$\frac{|A - B|(A + B)}{b^2} \leq \frac{2\|\Gamma_t\|\|\Gamma_t^\mathsf{T}(z - w)\|_2}{\sqrt{\lambda}} \leq 2\|\Gamma_t\|^2 \frac{\|V_t^{1/2}\|}{\sqrt{\lambda}} \epsilon.$$

Putting things together, we have
$$|R_t(u) - R_t(v)| \leq \frac{4\|\Gamma_t\|^2 \|V_t^{1/2}\|}{m\sqrt{\lambda}} \epsilon. \qquad\square$$

# G Proof of Lemma 12: concentration result

This proof will require the following definition of conditional subgaussianity, and the standard de la Peña-type concentration result for sequences of such random variables, stated thereafter.

**Definition 22.** *Let $A, B$ be random variables and $\mathcal{F}$ a $\sigma$-algebra. We say that $A$ is $\mathcal{F}$-conditionally $B$-subgaussian when $B$ is non-negative and $\mathcal{F}$-measurable, and for all $s \in \mathbb{R}$,*

$$\mathbb{E}[\exp\{sA\} \mid \mathcal{F}] \leq \exp\{s^2 B^2/2\} \quad \textit{almost surely.}$$

**Lemma 23.** *Let $(A_i, B_i, \mathcal{H}_i)_i$ be such that each $A_i$ is $\mathcal{H}_i$-conditionally $B_i$-subgaussian. Then, for any $x, y > 0$,*

$$\mathbb{P}\left\{\exists \tau \in \mathbb{N}: \left(\sum_{i=1}^{\tau} A_i\right)^2 \geq \left(\sum_{i=1}^{\tau} B_i^2 + y\right)\left(x + \log\left(1 + \frac{1}{y}\sum_{i=1}^{\tau} B_i^2\right)\right)\right\} \leq e^{-x/2}.$$

Lemma 23 is an immediate consequence of Theorem 2.1 in de la Pena et al. (2004), in particular comparing our Definition 22 with their condition (1.4). For more background, see Lattimore and Szepesvári (2020), Lemma 20.2, and the surrounding discussion.

We will also need the following three claims, verified in Appendix H. Therein,

$$\sigma_t = \sqrt{2Q_t^2 + Q_t R_t}.$$

**Claim 24.** *Each $W_{t+1}$ is $\sigma(\mathcal{F}_t \cup \mathcal{A}_t'')$-conditionally $\sigma_t/\sqrt{m}$-subgaussian.*

**Claim 25.** *For any $0 \leq t \leq \tau < T$, $1 + \sum_{i=\tau}^{t} \sigma_i^2 \leq 3 + \sum_{i=\tau}^{t} Q_i R_i$.*

**Claim 26.** *For any $0 \leq t \leq \tau < T$, on the event $\{R_0 \leq 2\}$, $1 + \sum_{i=\tau}^{t} \sigma_i^2 \leq (3 + 2T)^2$.*

*Proof of Lemma 12.* For any $\tau < T$, by Claim 24 and Lemma 23 applied with $y = 1/m$ and any $x > 0$, with probability $1 - \exp\{-\frac{x}{2}\}$, for all $t \in \{\tau, \ldots, T-1\}$,

$$\left(\sum_{i=\tau}^{t} W_{i+1}\right)^2 \leq \left(\frac{x}{m} + \frac{1}{m}\log\left(\sum_{i=\tau}^{t} \sigma_t^2 + 1\right)\right)\left(\sum_{i=1}^{\tau} \sigma_i^2 + 1\right).$$

We take $x = 2\log(3 + 2T)$, which by Claim 26 exceeds $\log(\sum_{i=\tau}^{t} \sigma_t^2 + 1)$ on $\{R_0 \leq 2\}$. Therefore, on the intersection of $\{R_0 \leq 2\}$ and the event implicitly defined above,

$$\left|\sum_{i=\tau}^{t} W_{i+1}\right| \leq \sqrt{\frac{2x}{m}}\sqrt{\sum_{i=1}^{\tau} \sigma_i^2 + 1}.$$

By our assumption on $m$, $\sqrt{2x/m} \leq 1/10$. Also, and using a trivial bound and Claim 25,

$$\sqrt{\sum_{i=1}^{\tau} \sigma_i^2 + 1} \leq \sum_{i=1}^{\tau} \sigma_i^2 + 1 \leq 3 + \sum_{i=\tau}^{t} Q_i R_i,$$

which combined with the display above yields the form of the claimed bound.

To conclude the proof, we take a union bound over $\tau \in \{0, \ldots, T-1\}$ and note that, by our assumption on $m$ and choice of $x$, $x/2 \leq m/400$. $\qquad\square$

# H Proofs of Claims 11 and 24 to 26

Let $\mathbb{F}_t'' = \sigma(\mathcal{F}_t \cup \mathcal{A}_t'' \cup \sigma(X_{t+1}))$, where we recall that $\mathcal{A}_t'' = \sigma(\mathcal{A}_t \cup \sigma(\xi_{t+1}, J_{t+1}, X_{t+1}))$ for each $t \in \mathbb{N}$, that $\mathbb{E}_t''$ denotes the $\mathbb{F}_t''$-conditional expectation, and that the purpose of conditioning on $\mathbb{F}_t''$ will be to integrate out the random targets $U_{t+1}^1, \ldots, U_{t+1}^m$.

*Proof of Claim 11.* Fix $u \in \mathbb{S}_2^{d-1}$ and note that

$$R_{t+1}^j = \frac{\langle u, S_t^j + U_{t+1}^j X_{t+1} \rangle^2}{\|u\|_{V_{t+1}}^2} = \frac{\langle u, S_t^j \rangle^2 + (U_{t+1}^j)^2 \langle u, X_{t+1} \rangle^2 + 2U_{t+1}^j \langle u, S_t^j \rangle \langle u, X_{t+1} \rangle}{\|u\|_{V_t}^2 + \langle u, X_{t+1} \rangle^2}. \tag{14}$$

Observe that $X_{t+1}$ and $S_t^j = S_0^j + \sum_{s \le t} U_s^j X_s$ are $\mathbb{F}_t''$-measurable and that $U_{t+1}^j$ is independent of $\mathbb{F}_t''$. The latter of these gives $\mathbb{E}_t'' U_{t+1}^j = 0$ and $\mathbb{E}_t'' (U_{t+1}^j)^2 = \frac{2}{3}$. With that, we have that

$$\begin{aligned}
\mathbb{E}_t'' R_{t+1}^j - R_t^j &= \frac{\langle u, S_t^j \rangle^2 + \frac{2}{3} \langle u, X_{t+1} \rangle^2}{\|u\|_{V_t}^2 + \langle u, X_{t+1} \rangle^2} - \frac{\langle u, S_t^j \rangle^2}{\|u\|_{V_t}^2} \\
&= \frac{\frac{2}{3} \langle u, X_{t+1} \rangle^2 \|u\|_{V_t}^2 - \langle u, S_t^j \rangle^2 \langle u, X_{t+1} \rangle^2}{\|u\|_{V_t}^2 \left( \|u\|_{V_t}^2 + \langle u, X_{t+1} \rangle^2 \right)} \\
&= \frac{\langle u, X_{t+1} \rangle^2}{\|u\|_{V_t}^2 + \langle u, X_{t+1} \rangle^2} \left( \frac{2}{3} - \frac{\langle u, S_t^j \rangle^2}{\|u\|_{V_t}^2} \right) \\
&= Q_t \left( \frac{2}{3} - R_t^j \right).
\end{aligned} \tag{15}$$

The statement follows by averaging over $j \in \{1, \dots, m\}$. □

*Proof of Claim 24.* Subtracting the first expression on the right-hand side of equation (15) from equation (14) and averaging over $j \in \{1, \dots, m\}$, we see that

$$W_{t+1} = R_{t+1} - \mathbb{E}_t'' R_{t+1} = \frac{Q_t}{m} \sum_{j=1}^{m} \left( (U_{t+1}^j)^2 - \frac{2}{3} \right) + \frac{1}{m} \sum_{j=1}^{m} U_{t+1}^j H_t^j$$

where $H_t^j = \langle u, X_{t+1} \rangle \langle u, S_t^j \rangle / \|u\|_{V_{t+1}}^2$. Note that $Q_t$ and $H_t$ are $\mathbb{F}_t''$ measurable and that $U_{t+1}^1, \dots, U_{t+1}^m$ are independent of $\mathbb{F}_t''$ and one another, and that their absolute values are bounded by 1. Thus, examining the two terms in the sum we see that:

- $\frac{Q_t}{m} \sum_{j=1}^{m} ((U_{t+1}^j)^2 - \frac{2}{3})$ is $\mathbb{F}_t''$-conditionally $\frac{Q_t}{\sqrt{m}}$-subgaussian.

- $\frac{1}{m} \sum_{j=1}^{m} U_{t+1}^j H_t^j$ is $\mathbb{F}_t''$-conditionally $\frac{H_t}{\sqrt{2m}}$-subgaussian, where

$$(H_t)^2 := \frac{1}{m} \sum_{j=1}^{m} (H_t^j)^2 = \frac{1}{m} \sum_{j=1}^{m} \frac{\langle u, X_{t+1} \rangle^2 \langle u, S_t^j \rangle^2}{\|u\|_{V_{t+1}}^4} = \frac{Q_t}{m} \sum_{j=1}^{m} \frac{\langle u, S_t^j \rangle^2}{\|u\|_{V_{t+1}}^2} = Q_t R_t.$$

The result follows by recalling that the sum of an $a$-subgaussian random variable and a $b$-subgaussian random variable is $\sqrt{2(a^2 + b^2)}$-subgaussian. □

The proof of the final two claims will require the following simple lemma.

**Lemma 27.** *Let $b_1, b_2, \dots$ be a sequence of real numbers in $[0, 1]$. Then, for any $\lambda > 0$ and $n \in \mathbb{N}^+$,*

$$\sum_{j=1}^{n} \frac{b_j}{\lambda + \sum_{i=1}^{j} b_i} \le \log(1 + n/\lambda) \quad and \quad \sum_{j=1}^{n} \left( \frac{b_j}{\lambda + \sum_{i=1}^{j} b_i} \right)^2 \le \frac{n}{\lambda(\lambda + n)} \le \frac{1}{\lambda}.$$

*Proof.* Let $B_0 = 0$ and for $j \in [n]$ let $B_j = B_{j-1} + b_j$. The function $f(x) = \frac{1}{\lambda + x}$ is decreasing on $[0, \infty)$. Hence, $\int_{B_{j-1}}^{B_j} f(x) dx \ge b_j f(B_j)$. Summing these up,

$$[\log(\lambda + x)]_0^{B_n} = \int_0^{B_n} f(x) dx = \sum_{j=1}^{n} \int_{B_{j-1}}^{B_j} f(x) dx \ge \sum_{j=1}^{n} b_j f(B_j) = \sum_{j=1}^{n} \frac{b_j}{\lambda + \sum_{i=1}^{j} b_i}.$$

Evaluating the left-hand side and noting that $B_n \leq n$ holds because $b_i \leq 1$ gives the first result. For the second sum, we use a similar argument with $g(x) = 1/(\lambda + x)^2$:

$$
\begin{aligned}
\sum_{j=1}^n \left( \frac{b_j}{\lambda + \sum_{i=1}^j b_i} \right)^2 &= \sum_{j=1}^n \frac{b_j^2}{(\lambda + B_j)^2} \\
&\leq \sum_{j=1}^n \frac{b_j}{(\lambda + B_j)^2} && \text{(because } 0 \leq b_j \leq 1) \\
&\leq \int_0^n g(x) dx && (g \text{ is decreasing on } [0, \infty), B_n \leq n) \\
&= \left[ \frac{-1}{\lambda + x} \right]_0^n = \frac{n}{\lambda(\lambda + n)} . && \square
\end{aligned}
$$

*Proof of Claim 25.* Noting that since $\|u\|_2 = 1$ and $\lambda \geq 5$, by Lemma 27,

$$
\sum_{i=\tau}^t Q_i \leq \sum_{i=0}^{T-1} \frac{\langle u, X_{i+1} \rangle^2}{\lambda + \sum_{j=0}^{i+1} \langle u, X_j \rangle^2} \leq \log(1 + T/5) \leq T
$$

and

$$
\sum_{i=\tau}^t Q_i^2 \leq \sum_{i=0}^{T-1} Q_i^2 = \sum_{i=0}^{T-1} \left( \frac{\langle u, X_{i+1} \rangle^2}{\lambda + \sum_{j=0}^{i+1} \langle u, X_j \rangle^2} \right)^2 \leq \frac{1}{\lambda} \leq 1 .
$$

Using these, we have

$$
1 + \sum_{i=\tau}^t \sigma_i^2 = 1 + 2 \sum_{i=\tau}^t Q_i^2 + \sum_{i=\tau}^t R_i Q_i \leq 3 + \sum_{i=\tau}^t R_i Q_i \leq 3 + T \max_{\tau \leq i \leq t} R_i . \qquad \square
$$

*Proof of Claim 26.* Since $(a + b)^2 \leq 2a^2 + 2b^2$ and by symmetry,

$$
\begin{aligned}
R_i^j = \frac{\langle u, S_0^j + \sum_{i=1}^i U_\ell^j X_\ell \rangle^2}{\lambda + \sum_{\ell=1}^i \langle u, X_\ell \rangle^2} &\leq 2R_0^j + 2 \frac{\left( \sum_{\ell=1}^i \langle u, X_\ell \rangle \right)^2}{\lambda + \sum_{\ell=1}^i \langle u, X_\ell \rangle^2} \leq 2R_0^j + 2 \max_{b \in [0,1]} \frac{(ib)^2}{\lambda + ib^2} \\
&\leq 2R_0^j + 2i.
\end{aligned}
$$

By definition, $R_i = \frac{1}{m} \sum_{j=1}^m R_i^j$, and by assumption $R_0 \leq 2$ and $i \leq T-1$, so $R_i \leq 4 + 2i \leq 2 + 2T$. And so,

$$
3 + T \max_{\tau \leq i \leq t} R_i \leq 3 + T(2 + 2T) \leq (3 + 2T)^2 . \qquad \square
$$

