# OpenReview forum: "Ensemble sampling for linear bandits: small ensembles suffice"
_NeurIPS.cc/2024/Conference — NeurIPS 2024 poster_

### Official Review · Reviewer_YgVw · 2024-07-09

**Soundness:** 3
**Presentation:** 3
**Contribution:** 3
**Rating:** 7
**Confidence:** 3

**Summary:**

This paper presents theoretical results for an exploration algorithm using ensemble sampling in the stochastic linear bandit setting. The proposed algorithm creates $2m = O(d \log T)$ ensemble models, selects one model uniformly at random, and then chooses the greedy action based on the parameters of that model.
The proposed algorithm achieves a regret bound of $\tilde{O}(d^{5/2} \sqrt{T})$, and it does not depend on the size of the action set.

**Strengths:**

- Unlike previous ensemble sampling algorithms (Lu & Van Roy, 2017; Qin et al., 2022), the proposed algorithm presents a frequentist regret analysis with a reasonable ensemble size. The authors also explain well the differences (e.g., symmetrization) compared to existing randomized algorithms like Thompson sampling and perturbed history exploration.
- The master theorem (Theorem 1) for analyzing the regret bound of the randomized algorithm using the probability that the selected parameter by the algorithm is more optimistic than the true parameter is also intriguing. Proof seems correct, even though I didn’t check very carefully.

**Weaknesses:**

- As noted in Remark 4, although this paper provides a meaningful theoretical result for ensemble sampling, compared to randomized algorithms like Thompson sampling and perturbed history exploration, the proposed algorithm is neither computationally efficient nor statistically efficient. Ensemble sampling is generally effective in complex settings such as DeepRL. However, the regret bound presented in this paper does not seem sufficient to prove the validity of the ensemble algorithm even in the linear setting.
- Thompson sampling algorithms generally have looser regret bounds than UCB algorithms but are known to perform better empirically. It would be beneficial if there were experimental results showing that ensemble sampling algorithms could empirically outperform existing randomized algorithms.

**Questions:**

1. When I first saw the ensemble sampling setting, it reminded me of the LSVI-PHE algorithm for linear MDPs with perturbed history exploration (Ishfaq et al., 2021). Ishfaq et al. (2021) address algorithms in the MDP setting, but with a horizon length of 1 ($H=1$), it can align with the linear bandit setting. Ishfaq et al. (2021) estimate perturbed parameters using Gaussian noise on observed rewards from previous episodes, repeating this process $M$ times. Actions are chosen based on the most optimistic of these estimated parameters. If we consider each perturbed estimator as an ensemble model, we have an ensemble size of $M$. However, the algorithm in Ishfaq et al. guarantees a high probability regret bound of $\tilde{O}(d^{3/2} \sqrt{T})$ (when $H=1$), with a theoretical sampling size of $M=O(d)$.
The only difference from the proposed ensemble sampling in this paper is that this paper selects the ensemble model to play uniformly at random. If the most optimistic model were chosen as in Ishfaq et al., could a better regret bound be guaranteed? Additionally, are there any other differences between the algorithm in Ishfaq et al. and the one proposed in this paper?

* * *
Ishfaq, Haque, et al. "Randomized exploration in reinforcement learning with general value function approximation." International Conference on Machine Learning. PMLR, 2021.

**Limitations:**

The authors have addressed the limitations in Section 4.
The content discussed in this paper appears to have little to no negative societal impact.

---

> ### Author Rebuttal · Authors · 2024-08-06
>
> Thank you for your review. There is an unfortunate, but crucial, misunderstanding, in the relation between our work and algorithms such as Thompson sampling and Perturbed History Exploration (PHE), which we believe is responsible for your low scoring of our work. We have clarified these relations in our main rebuttal, which we would ask you to consider.
>
> With the contents of the main rebuttal in mind, let’s discuss LSVI-PHE. Yes, LSVI-PHE fits an ensemble of $m$ estimates, just like we do. But, being an instance of PHE, it fits $m$ estimates from scratch at every single iteration! In contrast, Ensemble Sampling only updates each of the $m$ ensemble elements with the new received observation—so ensemble sampling does something akin to a single gradient step for each of the $m$ estimates at each timestep, whereas LSVI-PHE trains $m$ models to convergence at each step. This is a huge difference in the practicality of the methods, and the reason ensemble sampling is so interesting.
>
> As to taking a maximum over the ensemble: this is a very good question. Yes, you could do that with our iterative updates, and you’d get a better regret bound; you would recover the regret of Thompson sampling (so, $\sqrt{d}$ loose on LinUCB). But it is completely unclear that this would give a better algorithm: after all, it is random-element-type algorithms that are used in practice (see Osband et al. 2016, Osband et al. 2018 and papers that cite these), and practitioners are pretty good at trying out different sensible alternatives. We thank the reviewer for this suggestion. We will insert a comment discussing this point in the manuscript, acknowledging the reviewer.
>
> Regarding how the proof works for the version where the maximum is taken: Whether taking a random element or the maximum, the technical challenge in getting a bound is the same: it is showing that there exists at least one optimistic ensemble element in each “direction” of the parameter space when incremental updates are used. This result and the techniques we use to show it are the main contributions of our paper.
>
> We hope that the above and our main rebuttal clarified the key aspects of the paper. We will use the additional page available at the camera ready stage to clarify these points to future readers.
>
> With that said, we must emphasise that our paper solves a long standing open problem, which attracted two failed attempts previously published at NeurIPS. We believe that a well-written, sound paper with novel proof ideas and techniques that solves a hard open problem—one clearly of interest to the community—is a strong contribution to NeurIPS. As such, we would ask the reviewer to reconsider their assessment of our manuscript.
>
> _Osband, Ian, et al. "Deep exploration via bootstrapped DQN." Advances in neural information processing systems 29 (2016)._
>
> _Osband, Ian, John Aslanides, and Albin Cassirer. "Randomized prior functions for deep reinforcement learning." Advances in Neural Information Processing Systems 31 (2018)._

---

> ### Author Response · Authors · 2024-08-11
>
> Dear reviewer,
>
> The author-reviewer discussion period is shortly coming to an end. We were hoping to get a confirmation from you that you've considered our rebuttal, and to let us know if you have any further questions.

---

> > ### Comment · Reviewer_YgVw · 2024-08-12
> >
> > I appreciate the authors for the detailed explanation. Now, I understand the difference between ensemble sampling and PHE better. It would be great to include this explanation in the main text. However, according to the author’s explanation, PHE does not seem to be equivalent to Thompson sampling, at least in the linear case, from the perspective that PHE requires refreshing the noise for observations from previous episodes in each round. Note that in LinTS (Agrawal & Goyal, 2013), the mean vector and covariance matrix are updated incrementally (i.e., $\mathbf{V}\_t = \lambda \mathbf{I}\_d + \sum\_{i=1}^t X\_i X\_i^\top = \mathbf{V}\_{t-1} + X\_t X\_t^\top$, $\mathbf{b}\_t = \sum\_{i=1}^t Y\_i X\_i = \mathbf{b}\_{t-1} + Y\_t X\_t$ ).
> >
> > Now consider a variant of Thompson sampling where sampling $M$ parameters $\theta\_t^{(j)} \sim \mathcal{N}(\mathbf{V}\_{t-1}^{-1} \mathbf{b}\_t,   \nu \mathbf{V}\_t^{-1})$ for $j \in [M]$, and select an action based on the most optimistic parameter, i.e., $X_t = \arg \max\_{x, j} \langle x, \theta_t^{(j)} \rangle$ (or we may choose a parameter uniformly random as this paper does). This variant of Thompson sampling would guarantee the same $\tilde{O}(d^{3/2} \sqrt{T})$ regret as traditional TS (Agarawal & Goyal, 2013), but it also can work online. It would be helpful if you could explain the advantages of the ensemble approach in this paper compared to this variant of Thompson sampling.
> >
> > Moreover, as the author mentioned, although this paper theoretically addresses a longstanding open problem that had previously been a failed attempt published in NeurIPS, I do not believe the paper offers a complete solution to the open problem.
> >
> > Certainly, ensemble sampling has shown to be tractable and empirically effective for nonlinear, complex tasks like Deep RL. Given this, one would intuitively expect that ensemble sampling would be at least equivalent to or provide a tighter regret bound than existing randomized exploration algorithms, even in a linear setting. However, I believe the theoretical results proposed in this paper, while very interesting, are only a partial solution.
> >
> > According to the rebuttal to Reviewer Pgmx, the author claims that “if an algorithm provably cannot work in the linear setting, chances are it won’t work beyond it, either”. However, by the same token, the results of this paper indicate that ensemble sampling is not optimal even in the linear setting. Furthermore, if ensemble sampling is advantageous only when a closed form is not provided, wouldn’t there be no reason to use ensemble sampling in linear settings where a closed form is given? Although the paper presents a novel proof technique, I do not think it offers a solution to the longstanding open problem, especially since even this technique, which is based on the closed-form of the linear setting, fails to demonstrate the efficacy of ensemble sampling in the linear setting where a closed form is provided.

---

> > > ### Comment · Area_Chair_oCxT · 2024-08-12
> > >
> > > Thank you to both the authors and the reviewers for the excellent discussion here. I would like to chime in briefly regarding the theoretical and empirical merits of randomized exploration techniques. According to Proposition 3.2 of Ishfaq et al. (2021), LSVI-PHE should be equivalent to Thompson Sampling in the linear case. However, the concept of PHE is more easily generalizable to complex settings where the reward is nonlinear (Kveton et al., 2020; Jia et al., 2022).
> > >
> > > Regarding the practical use of randomized exploration, I believe that recently proposed algorithms, such as LMC-TS (Xu et al., 2022) and LSVI-LMC (Ishfaq et al., 2023), which utilize MCMC to approximate complex posterior distributions, serve as strong baselines. These algorithms not only achieve the same regret as TS algorithms but also feature simple algorithmic designs that generalize well to nonlinear and complex environments.
> > >
> > > Therefore, it would be beneficial to discuss in detail the pros and cons of the algorithm studied in this paper compared with those randomized methods in the literature, both empirically and theoretically.
> > >
> > > **References**
> > >
> > > Ishfaq, Haque, et al. "Provable and practical: Efficient exploration in reinforcement learning via langevin monte carlo." arXiv preprint arXiv:2305.18246 (2023).
> > >
> > > Jia, Yiling, et al. "Learning Contextual Bandits Through Perturbed Rewards." arXiv preprint arXiv:2201.09910 (2022).
> > >
> > > Kveton, Branislav, et al. "Randomized exploration in generalized linear bandits." International Conference on Artificial Intelligence and Statistics. PMLR, 2020.
> > >
> > > Xu et al. “Langevin monte carlo for contextual bandits.” International Conference on Machine Learning (2022).

---

> ### Author Response · Authors · 2024-08-12
>
> __Clarifying the confusion about the goal of studying ES in the linear setting__ A minimum requirement for an algorithm that aims at some level of generality is that it should also work reasonably well in simpler settings. Moreover, if the algorithm achieves some generality, compromising on optimality in a simpler setting (in our case, the linear setting) should be  acceptable. We are sure that the reviewer is also aware of the many examples where such a compromise exist or is even unavoidable (there is a large body of literature on problems of this type). Ours is the first paper that gives *any* sort of theoretical support for ES. Given the amount of work devoted to ES (empirical and theoretical), we think that a paper that changes this poor status quo should be of major interest (news!) to the community. We note in passing that all previous papers on analyzing ES started with the linear setting (Lu and Van Roy'17; Phan et al,'19; Qin et al,'22 -- all papers published at NeurIPS!), and their rationale for studying ES in the ensemble sampling was exactly the same as ours (see, e.g., the introduction of the paper by Qin et al.)
>
> Another confusion that we would like to clarify is that our work does not give a definite answer to whether ES can be optimal (or even meet the guarantee TS can). While we could not obtain a result like this, ours is an *exponential* improvement over the previous result in terms of what ensemble size gives the optimal rate of growth of regret (up to logarithmic factors).
>
> While we also like to think that we have high standards, expecting that every paper will close a whole area of study by providing a final answer is unrealistic and unhealthy for the community. Just to mention an example close to our work, by this standard, the LinTS work of Agrawal & Goyal (2013) also cited and perhaps liked by the reviewer, which we think is a breakthrough paper, should not have been published: As is well known, the regret bound given in this work falls short of that available for LinUCB by a multiplicative factor of size $\sqrt{d}$. Yet, this paper sparked a lot of good work in the community. Today, we even know that the extra $\sqrt{d}$ factor is even unavoidable (Hamidi et al'20). But achieving this result required the cooperation of many researchers through many years, which was greatly facilitated by reviewers who let the many papers that provided some small steps towards the final solutions to be published. In light of this, we respectfully ask the reviewer to reconsider their position as to what constitutes an interesting, publishable result.
>
> Finally, at the risk of repeating earlier arguments, we would like to note that even if in the linear setting ES is inferior to alternatives, the alternatives loose their edge against ES as soon as the setting becomes slightly more complicated. We show this now in the deep learning setting.
>
> __On ES vs PHE in Deep Learning__ Suppose that our model is a neural network with $p$ parameters, and we use a standard NTK-style linearisation (see, e.g., Jia et al., 2022). TS/PHE with incremental updates gives:
> - $O(p^2)$ per step computational complexity for the matrix-vector product, used for Sherman-Morrison updates
> -  $O(p^2)$ memory for storing the covariance matrix.
>
> Consider on the other hand Ensemble Sampling in the NTK setting. Let $d$ denote the effective dimension of the RKHS with feature map given by the gradient of the neural network. Then the required ensemble size will be $m=d \log T$ (crucially, not $p \log T$). Updating each ensemble element requires a step (or a number of steps) of backpropagation, so order $O(p)$ runtime and $O(p)$ memory. With that, ensemble sampling uses
> - $O(dp \log T)$ runtime,
> - $O(dp \log T)$ memory.
>
> Observe that $d\leq p$ always. In cases where $d$ might be much smaller than $p$, ensemble sampling is beneficial. __This is why ES works in the deep learning setting, and TS/PHE does not.__
>
> One might then say: okay, why don't we implement TS/PHE with sketching? That will make the runtime of TS depend on $d$ in place of $p$. But implementing sketching for large models is _very difficult,_ due to both numerical stability issues and choosing parameters involved in sketching. We haven't seen popular practical methods built around sketching.
>
> We will be happy to include some extra discussion of why ensemble sampling is a sensible choice if the paper gets accepted using the extra page. This seems something easy to address would the reviewer think this to improve the paper.
>
> In conclusion, we are not claiming that we solve all longstanding open problems related to ES, or even that we solve the hardest of these, but we claim that we solve some of these problems, which we think are of major interest to the community, especially given the new proof techniques we employ.

---

> > ### Comment · Reviewer_YgVw · 2024-08-14
> >
> > Thank you for the author's detailed response. I understand the motivation behind ES research in the linear setting and believe that the contributions of this paper should be appreciated. (Please note that I support the acceptance of this paper.) However, I feel that the current manuscript and the author rebuttal do not fully explain that motivation, and I believe this aspect needs to be supplemented.
> >
> > I am not trying to depreciate this work; rather, I believe the results in this paper can be further strengthened through appropriate comparisons and motivation with other works.
> >
> > I sincerely hope that the content of the author-reviewer discussion has helped the authors to further polish their work.
> >
> > Accordingly, I will raise my score from 5 to 7.

---

> ### Author Response · Authors · 2024-08-12
>
> Thank you for the comment. We will be happy to expand the discussion of randomized methods, as indicated beforehand.
>
> We agree with the AC on the motivations for using PHE. And if one thinks of PHE, running ES instead is a natural choice. We prove that doing so still achieves reasonable regret–-even if our result is not optimal (or maybe it is, in the sense that maybe there is a real cost to ES over PHE/TS, and a tighter result is not possible).
>
> On "discussing in detail the pros and cons of the algorithm studied in this paper compared with those randomized methods in the literature":
> As confirmed in our rebuttal, we would be happy to spend the additional page granted at camera ready on this discussion. We could also include a more extensive related works section in the appendices. Nonetheless, we must emphasize that our novel contribution is theoretical, not algorithmic–-we do not introduce ensemble sampling, its already a commonly used method in the applied literature–-and there is a strict page limit. We are also not writing a tutorial paper on randomized methods, but providing a novel theoretical contribution. We are writing for an expert that understands randomized methods, and wants to know what specific tricks are required to show that ensemble sampling works: for an expert that can make the comparison between methods themselves. Nevertheless, again, we are happy to use the extra space for a discussion of the relative merits of all methods mentioned.

---

### Official Review · Reviewer_N2aY · 2024-07-11

**Soundness:** 3
**Presentation:** 3
**Contribution:** 4
**Rating:** 7
**Confidence:** 4

**Summary:**

The paper presents a regret analysis of ensemble sampling within the stochastic linear bandit framework. It demonstrates that an ensemble size scaling logarithmically with time and linearly with the number of features suffices, marking a theoretical advancement. The paper shows that under standard assumptions for a \(d\)-dimensional stochastic linear bandit with an interaction horizon \(T\), the regret is of the order \((d \log T)^{5/2} \sqrt{T}\). Although the regret bound is not as tight as the existing bounds for the linear bandits, this is the first analysis of ensemble sampling in a structured setting that avoids the need for the ensemble size to scale linearly with \(\sqrt{T}\), which previously compromised the purpose of ensemble sampling, while still achieving near-optimal order regret.

**Strengths:**

- The paper makes a decent theoretical contribution by reducing the required scaling of ensemble size, which enhances the applicability and efficiency of ensemble sampling in linear bandit problems. By showing the better scaling of ensemble size, the paper provides a pathway for more efficient implementation of the ensemble sampling algorithms.
- To my knowledge, this is the first correct frequentist regret analysis of linear ensemble sampling.

**Weaknesses:**

- One drawback is that, as the authors acknowledge, the regret bound of \(O(d^{5/2} \sqrt{T})\) is relatively loose compared to existing linear bandit results. For example, TS-based algorithms like LinTS typically achieve \(O(d^{3/2} \sqrt{T})\) regret, making the proposed bound clearly suboptimal.
- The dependence on \(m\) in the regret bound indicates that as the ensemble size increases, the algorithm's regret performance worsens. This superlinear dependence, with regret being \(O(m^{3/2})\), raises concerns. Given this result, it is unclear why one would opt for this particular way (as proposed in the paper) of linear ensemble sampling.

**Minor comment (but for clarity)**
- I recommend that the authors explicitly write the linear \(d\)-dependence on the ensemble size \(m\) instead of shoving \(d\)-dependence into \(N\) in the theorem statement of Theorem 1 if possible, as I see that authors are transparent about $d \log T$ dependence on $m$ in Remark 1.
- I am not sure whether the expression "**slightly** worse than that obtained for Thompson sampling" (in Line 120) is adequate given that such a gap (extra $d$) can be seen as **significantly** by among many bandit researchers. I understand the authors' intention, but I suggest removing such an expression of "slightly."

**Questions:**

- Regarding the comments in Remark 3, under unknown $T$, can't you derive some regret bound with the doubling epoch trick though? Or, do you argue that you cannot get any bound (sublinear in T) even with the doubling trick?

**Limitations:**

There is no separate "Limitations" section. But the authors discuss the weakness of the proposed method.

---

> ### Author Rebuttal · Authors · 2024-08-06
>
> We must clarify two important misunderstandings:
> 1. Our upper bound scales superlinearly with m, but that does not mean that the regret of the algorithm scales superlinearly with m. Our bound is simply not tight when m goes to infinity (we will adjust the wording of Remark 2 to make this clear). It is tight for small ensembles, which is the crucial regime (we do not think anyone has any doubts whether ensemble sampling can work when m goes to infinity! Even previous results of Qin et al. 2022 show that it can, at least in the Bayesian set-up). Our interest is in showing that ensemble sampling as it is used in practice, with small ensembles, can work; and that’s what we demonstrate.
> 2. Our algorithm is not a “particular version” of ensemble sampling. It's almost the standard version, with the only requirement being a symmetrisation of the ensemble, which can clearly only be beneficial in practice and should be done anyway. In any case, we have no doubt that our result would hold without symmetrisation; but showing this would involve a much, much more tedious argument, from which it would be significantly harder to extract useful insight on how and why ensemble sampling actually works. We address this point in remark 9 of our paper.
>
> We are happy to correct the two issues raised as “minor comments” in the way suggested by the reviewer.
>
> Regarding remark 3: yes, one can of course use the “doubling trick” to obtain an algorithm the regret bound of which is only a constant larger. The problem is that the doubling trick doesn’t just increase the regret bound (which is harmless) by some factor, but it will increase the actual regret incurred by said factor (which is bad). The aim of our paper is to analyse the algorithm as practitioners might use it, and for the aforementioned reason, practitioners would use the doubling trick only as a last resort if ever. Thus, implementing the algorithm without knowledge of the horizon in a way that does not increase the actual regret incurred much in comparison to if the horizon were known is a problem we do not have an answer to—and one where we suspect that a good answer might not exist.
>
> Regarding the weakness of our bound: yes the bound is likely suboptimal; but it is also exponentially better in two quantities than the previously given bound, and holds for the more stringent minimax regret criterion, rather than for Bayesian regret, as in the previous paper. We must point out that the previous paper with the much weaker guarantees was published at NeurIPS in 2022 (Qin et al. 2022), and therefore we believe that our much stronger results are more than sufficient for publication.
>
> Note that we also detail what it would take to get a bound better than ours in Remark 10 of our paper; the required arguments around order statistics in a sequential setting are far beyond the current literature on the topic, but there is hope that it might be solvable in the coming years—time uniform concentration bounds are currently a hot topic, and work on similar questions is beginning to pop up in the literature.
>
> In light of our rebuttal, we must ask the reviewer to reconsider their recommendation of only a “weak accept” for a paper that improves exponentially in multiple respects over a previously published NeurIPS work;  a paper that solves a rather long-standing problem in theory; a paper comprising many novel proof ideas and interesting techniques.
>
> _Qin, Chao, et al. "An analysis of ensemble sampling." Advances in Neural Information Processing Systems 35 (2022): 21602-21614._

---

> ### Comment · Reviewer_N2aY · 2024-08-09
>
> I think these results deserve recognition. I am raising my score from 6 to 7.

---

> > ### Author Response · Authors · 2024-08-11
> > **thanks**
> >
> > Thank you!

---

### Official Review · Reviewer_P6Xa · 2024-07-12

**Soundness:** 3
**Presentation:** 2
**Contribution:** 1
**Rating:** 6
**Confidence:** 2

**Summary:**

The authors study an upper confidence bound (UCB) type of ensemble sampling method specific to the stochastic linear bandits. Based on previous work, it improves the analysis by introducing Rademacher variables for symmetrising the ensemble. The authors are thus able to obtain $\sqrt{T}$ dependency in the regret with an ensemble size scaling with $d$ (dimension) and $\log T$ instead of $T$ (horizon).

**Strengths:**

- Removing the linear dependency of $m$ (ensemble size) on $T$ (horizon) for this ensemble sampling method is significant.
- The master theorem in Section 3 introduces intermediate filtrations $A'$ that may be of independent interest.

**Weaknesses:**

- The point of introducing the ensemble sampling in [Lu and Van Roy 2017] seems to be approximating Thompson sampling for complex models where computation is intractable. Meanwhile, the regret of linear bandits is known and achievable by LinUCB. Although the analysis improves for using smaller ensemble size in linear bandits, it deviates from the difficult setup and hence undermines the contribution of this paper.
- The number and omnipresence of remarks in the main text obscures the core idea of this paper. For example, Remark 3 does not add much to its section (the upper bound) and is under-explained.

**Questions:**

- I may have missed this - can the authors point out the significance of the regularization parameter λ? It seems from the theorems (2 and 5) λ only needs to be larger than 5 (i.e. an absolute constant)?

---

> ### Author Rebuttal · Authors · 2024-08-06
>
> We would ask that the reviewer confirms that they have read our main rebuttal, and to acknowledge that they understand and agree that our paper solves an open problem in the theory community that has attracted two previous failed attempts—both published at NeurIPS. With that in mind, we would ask the reviewer to reconsider their recommendation to reject our work: NeurIPS is precisely the place for well-written, sound papers that address hard open problems highlighted by the community with novel proof ideas and techniques.
>
> We would also ask the reviewer to clarify the issue with respect to remark 3: to us, the lack of a good anytime version of ensemble sampling poses an interesting problem. We would have also liked to see some more details of as to why the reviewer thinks that remark 3 is underexplained. We acknowledge that the text can be dense, but we ask the reviewer not to forget that we are bound by a length constraint, and we think the text has just enough information to understand the issue (here and elsewhere).
>
> As to the question regarding $\lambda$: the significance of $\lambda$ is that it is the scale of the noise injected into our estimates of the instance parameter prior to observing any data. The lower bound on lambda shows that it is crucial to inject some noise—this has already been understood by practitioners, and indeed the main contribution of Osband et al. 2018 over Osband et al. 2016 is the introduction of such “prior noise”. Our theory successfully confirms what practitioners have observed and developed an intuitive/heuristic understanding of (this intuitive understanding is outlined in Osband et al. 2019). This is a non-trivial contribution, and we will highlight it in our next revision of the manuscript.
>
> _Osband, Ian, et al. "Deep exploration via randomized value functions." Journal of Machine Learning Research 20.124 (2019): 1-62._

---

> > ### Comment · Reviewer_P6Xa · 2024-08-09
> >
> > I agree that the work can be extended with NTK and itself is of interest to the theoretical community. And it would be a good addition to provide the discussion on $\lambda$.
> > Given the placement of its previous attempts and the authors' response, I'll change my rating toward having this work published i.e. from 4 to 6.

---

### Official Review · Reviewer_Pgmx · 2024-07-12

**Soundness:** 3
**Presentation:** 3
**Contribution:** 2
**Rating:** 6
**Confidence:** 3

**Summary:**

This paper studies the ensemble sampling in the stochastic linear bandit setting, and shows that the ensemble sampling can achieve $\mathcal{O}( (d\log T)^{5/2}  \sqrt{T})$ regret with an ensemble size of order $\mathcal{O}(d log T)$. The authors claim this is the first meaningful result of ensemble sampling as it does not require sub-linear ensemble size.

**Strengths:**

1. Though I have skimmed the proof of several lemmas, the analysis part seems to be rigorous and mathematically correct. The notations are clear defined before their references.
2. The result which shows that ensemble learning is able to deal with linear bandit problem is quite interesting. This result would be applied in other similar settings.

**Weaknesses:**

1. It would be better to emphasis the differences between Algorithm 1 and LinUCB and explain how these differences work. The beta is quite similar.
2. Though the authors refer the audience to the previous works for the motivation of studying ensemble sampling, it's still beneficial to include the related works on linear bandits and briefly discuss the specific advantages of using ensemble sampling.

**Questions:**

The questions are raised in the weakness section. I am willing to re-evaluate the scores if these questions are properly answered.

**Limitations:**

This paper is pure theoretical and does not have any limitations.

---

> ### Author Rebuttal · Authors · 2024-08-06
>
> While our focus is on the novel proof ideas and techniques needed to get a bound for ensemble sampling, we would be happy to use the additional page available at the camera ready stage to provide more context on ensemble sampling and its relation to other methods.
>
> To answer the reviewer’s two “weaknesses” points directly:
> 1. The beta featuring in our work is indeed the same as that used in LinUCB, as it is the standard confidence width multiplier used for the confidence ellipsoids of linear ridge regression, as obtained from the method-of-mixtures. However, the difficulty with ensemble sampling lies in proving that the ensemble parameters are “sufficiently spread out” in the confidence ellipsoid in an appropriate sense. Thus, while we reuse part of the ideas from standard bandit proofs (optimism), much work had to go into showing that optimism thus indeed holds (in a certain sense).
> 2. Regarding advantages of using ensemble sampling, as discussed in Remark 4: Ensemble sampling should only be used in settings where closed forms are not available. There are no advantages to using ensemble sampling in the linear setting studied in this work, where Thompson sampling has a closed-form solution. The linear setting is studied in theoretical work because it provides a testing ground for the soundness of algorithms: if an algorithm provably cannot work in the linear setting, chances are it won’t work beyond it, either.
>
> With this out of the way, we ask the reviewer to consider our main rebuttal. In particular, we ask the reviewer to reconsider their assessment of our contribution as merely “fair”; we solve a long-standing open problem that has had two failed attempts published at NeurIPS. And we would ask that the reviewer reconsiders their scoring of the paper as “Borderline accept: Technically solid paper where reasons to accept outweigh reasons to reject”, since to our reading, the reviewer has not provided any significant reasons to reject, whereas reasons to accept are plentiful: a very strong theoretical contribution with interesting, novel proof techniques, on a question that has attracted much attention in the community.

---

> ### Author Response · Authors · 2024-08-11
>
> Dear reviewer,
>
> The author-reviewer discussion period is shortly coming to an end. We were hoping to get a confirmation from you that you've considered our rebuttal, and to let us know if you have any further questions.

---

> > ### Comment · Reviewer_Pgmx · 2024-08-13
> >
> > Sorry for the late reply.
> >
> > Thank the authors for their response. I would like to increase my score to 6.

---

### Author Rebuttal · Authors · 2024-08-06

We respectfully disagree with the current assessment of our work and would like to encourage the reviewers to reconsider the following: We believe that the successful resolution of a long-standing open problem, one that has attracted multiple prior attempts, holds significant value within the NeurIPS community. Notably, two of these prior attempts were themselves published at NeurIPS (Lu & Van Roy 2017 and Qin et al. 2022). A well-written and sound paper addressing this open problem is a significant contribution. Novel proof ideas and techniques are of interest and contribute to the advancement of the field. It appears to us that none of the reviewers considered these aspects of our work; hence, again, we respectfully ask the reviewers to reconsider their assessment taking the above into account. We will also make an effort to revise the paper to make it (even) more clear that the above is what makes our paper interesting for the community, along with some other revisions (which we think are minor) to clarify a few things that seemed to be missed by the reviewers; in particular, the relation of ensemble sampling and Thompson sampling/perturbed history exploration, and the applicability of our results in the deep learning setting.

__Relation to TS/PHE__ There was a common misunderstanding amongst the reviewers as to the relation between ensemble sampling (ES), Thompson sampling (TS) and Perturbed History Exploration (PHE). In the linear setting, PHE and Thompson sampling are equivalent, and so we will make the comparison only between Ensemble sampling (ES) and PHE. In PHE, at every round, you take all past observations, add fresh noise to all of the data, then fit a model, and select a greedy action. That means that at every iteration, you have to fit a new model! If the models are neural networks, this is extremely expensive.

Could you, perhaps, instead of fitting a new, fresh model at every round, have an ensemble of $m$ models, and update these incrementally? That is, at each round, you do not add fresh noise to all previous observations, but only to the observation just received. This is ensemble sampling; in contrast to PHE, it does not require fitting a new model from scratch at every round; and it does not require storing past data—it is a completely online algorithm.

This leaves the question of whether ES works; whether we can get away with using incremental updates, and not train models from scratch. This is the question that we resolve with an affirmative answer: yes you can! The answer to this question was genuinely not known at all before our work. Our regret bound isn’t as good as if you had trained $m$ new models each iteration… maybe our bound is suboptimal, or maybe it's a real trade-off: we do not know; we discuss this in Remarks 2, 4 and 10.

__Neural network function approximation__ The second concern of our reviewers was whether we had dodged the difficult “deep learning” setting by considering only the linear setting. We did not: the difficulty of proving a result for ensemble sampling comes from the incremental nature of the updates, and is tricky whenever there are any dependencies (correlations) between actions in the problem; the linear setting fully captures these difficulties. The core contribution of the paper is a completely novel proof addressing this difficulty. With a result on the regret proven for the linear setting, one can extend the result to deep learning using neural tangent kernel techniques—this is standard and requires no novel proof ideas or techniques; but at the same time rather tedious. Given the short page limit of NeurIPS papers, we decided not to include this in our manuscript. We believe that our contribution in the linear setting is valuable for its techniques and ideas, regardless of application to neural networks.

Furthermore, the difficulty is not in showing that an algorithm based on the linear bandit setting can have bounded regret when combined with neural network function approximation, but that it remains tractable in that setting. PHE and TS are not tractable in this setting. Ensemble sampling, on the other hand, is the method behind the BootstrapDQN and Ensemble+ algorithms of Osband et al. 2016 & Osband et al. 2018; as evidenced by the citation counts of these papers, these ensemble sampling algorithms are commonly used.

We are thus proving results for methods actually used in practice, rather than the PHE/TS algorithms, which while simple to analyse (and often analysed and extended in the theory literature), are not really used in deep reinforcement learning. We believe that proving results for algorithms that practitioners actually use, rather than for simpler theoretical constructs, is of crucial importance to the community, and often overlooked in theory work.

__Moving forward__ When writing the paper, given that PHE, Thompson sampling and ensemble sampling are not our inventions and are thoroughly described and discussed in the literature, we assumed that it was not necessary to describe the above mentioned points and that it was sufficient to refer to the literature; and that we should use the scarce space for describing the novel parts of our work instead. As the camera ready allows for an extra page, we now think that the extra space should be used to include a discussion of the relationship of these methods, making the motivation to analyse ensemble sampling as we do it, clear, regardless of the familiarity of the readers with the literature.

We must again emphasise that our paper solves a long-standing open problem, which attracted two failed attempts, both published at NeurIPS. We think that a well-written, sound paper that successfully solves this open problem is surely a strong contribution publishable at NeurIPS.

With the above in mind, we must ask that all reviewers reconsider their scoring of our paper.

---

### Author Response · Authors · 2024-08-07
**References**

Here are all the papers referred to in our rebuttals:

_Osband, Ian, et al. "Deep exploration via randomized value functions." Journal of Machine Learning Research 20.124 (2019): 1-62._

_Osband, Ian, et al. "Deep exploration via bootstrapped DQN." Advances in neural information processing systems 29 (2016)._

_Osband, Ian, John Aslanides, and Albin Cassirer. "Randomized prior functions for deep reinforcement learning." Advances in Neural Information Processing Systems 31 (2018)._

_Qin, Chao, et al. "An analysis of ensemble sampling." Advances in Neural Information Processing Systems 35 (2022): 21602-21614._

_Lu, Xiuyuan, and Benjamin Van Roy. "Ensemble sampling." Advances in neural information processing systems 30 (2017)._

---

### Comment · Area_Chair_oCxT · 2024-08-08
**Please read the authors’ rebuttal and reply by August 13, 2024 (11:59 PM, AOE time)**

Dear Reviewers,

Thank you for your hard work during the review process. The authors have responded to your initial reviews. **If you haven’t already done so, please take the time to carefully read their responses.** It is crucial for the authors to receive acknowledgment of your review by the deadline for the author-reviewer discussion period, which is August 13, 2024 (11:59 PM, Anywhere on Earth). Please address any points of disagreement with the authors as early as possible.

Best,

Your AC

---

### Decision · Program_Chairs · 2024-09-25

**Decision:**

Accept (poster)

**Comment:**

This paper provides the first frequentist regret analysis of ensemble sampling for linear contextual bandits. The authors present a $(d\log T)^{5/2}\sqrt{T}$ regret upper bound and only require an ensemble size of $d\log T$. Though the regret result is not optimal in linear bandits (and is larger than that of Thompson Sampling, PHE, and other randomized algorithms by a factor of $d$), ensemble sampling has been extensively adopted in the deep RL community. This work provides a valuable starting point for theoretically understanding ensemble sampling, which warrants its acceptance.

During the rebuttal and discussions, some reviewers raised questions regarding the motivation behind ensemble sampling, its theoretical analysis in the linear setting, and its comparison/discussion with other randomized exploration strategies. I hope the authors will address these comments and include the promised discussion in the camera-ready version.